# Mitochondrial copper and phosphate transporter specificity was defined early in the evolution of eukaryotes

Xinyu Zhu[1†], Aren Boulet[2†], Katherine M Buckley[1†], Casey B Phillips[1], Micah G Gammon[1], Laura E Oldfather[1], Stanley A Moore[2], Scot C Leary[2], Paul A Cobine[1*]

[1]Department of Biological Sciences, Auburn University, Auburn, United States; [2]Department of Biochemistry, Microbiology and Immunology, University of Saskatchewan, Saskatoon, Canada

**Abstract** The mitochondrial carrier family protein SLC25A3 transports both copper and phosphate in mammals, yet in *Saccharomyces cerevisiae* the transport of these substrates is partitioned across two paralogs: PIC2 and MIR1. To understand the ancestral state of copper and phosphate transport in mitochondria, we explored the evolutionary relationships of PIC2 and MIR1 orthologs across the eukaryotic tree of life. Phylogenetic analyses revealed that PIC2-like and MIR1-like orthologs are present in all major eukaryotic supergroups, indicating an ancient gene duplication created these paralogs. To link this phylogenetic signal to protein function, we used structural modeling and site-directed mutagenesis to identify residues involved in copper and phosphate transport. Based on these analyses, we generated an L175A variant of mouse SLC25A3 that retains the ability to transport copper but not phosphate. This work highlights the utility of using an evolutionary framework to uncover amino acids involved in substrate recognition by mitochondrial carrier family proteins.

*For correspondence:
paul.cobine@auburn.edu

†These authors contributed equally to this work

Competing interests: The authors declare that no competing interests exist.

## Introduction

Mitochondrial carrier family (MCF/SLC25) proteins comprise the largest family of mitochondrial inner membrane (IM) proteins and are responsible for transporting numerous substrates, including Krebs cycle intermediates, nucleoside di- and triphosphates for energy metabolism and nucleotide replication, amino acids for degradation or maintenance of the urea cycle, and essential metals such as copper (Cu) and iron (*Palmieri et al., 2020*; *Cunningham and Rutter, 2020*). Structurally, MCF transporters consist of a conserved fold with three repeats that contain two transmembrane helices connected by a short α-helical loop (*Robinson et al., 2008*; *Ruprecht and Kunji, 2020*). The repeated structural elements and variable copy numbers across eukaryotic phyla (53 in humans and 35 in yeast) suggest that this complex gene family arose through multiple duplication events followed by neofunctionalization as substrate needs changed. From an evolutionary perspective, one hypothesis is that protein families with multiple substrates (e.g., enzymes and transporters) arose as generalists that duplicated to evolve specificity over time (*Eick et al., 2017*; *Eick et al., 2012*). However, the evolutionary history of the MCF/SLC25 family with respect to substrate specificity remains largely unexplored.

Our current mechanistic understanding of MCF activity is based on in vitro transport assays, phenotypic observations made in mutant cells, and structures of the ADP-ATP carrier (*Palmieri, 2004*; *Ruprecht and Kunji, 2020*). This MCF transporter adopts two conformational states: the cytoplasmic, or c-state, which is open to the intermembrane space (IMS), and the matrix, or m-state, which is open to the matrix (*Ruprecht et al., 2019*; *Pebay-Peyroula et al., 2003*). All MCFs have six

transmembrane helices with conserved motifs that allow for formation of salt bridges and the close packing of helices that are critical to the mechanism of transport (*Ruprecht and Kunji, 2020*).

Cu is required in mitochondria for the stability and activity of the IM-embedded enzyme cytochrome *c* oxidase (COX) and the IMS-localized superoxide dismutase. The Cu used in the assembly of these enzymes comes from a pool in the mitochondrial matrix (*Baker et al., 2017*). We previously identified PIC2 as a mitochondrial Cu transporter in *Saccharomyces cerevisiae* (*Vest et al., 2013*). Mutant yeast strains lacking *PIC2* (*pic2Δ*) are deficient in COX activity and have lower mitochondrial Cu levels than isogenic wild-type (WT) strains (*Vest et al., 2013*). Silver (Ag$^+$) is an isoelectronic with Cu and has been widely used as a tool to interrogate Cu homeostasis (*Puchkova et al., 2019*). In yeast, inclusion of Ag$^+$ in the growth medium restricts Cu uptake and results in a mitochondrial Cu deficiency (*Vest et al., 2013*). This competition assay allowed us to identify yeast strains that could not overcome the Cu limitations imposed by Ag$^+$ when grown on non-fermentable carbon sources. We also exploited the toxicity of Ag$^+$ uptake to assay the Cu transport activity of these MCFs when expressed in *Lactococcus lactis* (*Boulet et al., 2018*; *Vest et al., 2016*; *Vest et al., 2013*). Expression of MCFs in *L. lactis* has been used extensively to assess transport activity because the proteins are readily expressed and inserted into the cytoplasmic membrane (*King et al., 2015*; *Monné et al., 2005*; *Kunji et al., 2005*; *Kunji et al., 2003*). The expression of a Cu-transporting MCF (e.g., PIC2) in *L. lactis* reduces the minimal inhibitory concentration of Ag$^+$ required for growth arrest.

Although PIC2 has also been implicated in phosphate transport (*Hamel et al., 2004*; *Takabatake et al., 2001*; *Fiermonte et al., 1998*; *Kolbe et al., 1984*), the primary phosphate-transporting MCF in yeast is MIR1 (*Phelps et al., 1991*; *Takabatake et al., 2001*). *PIC2* expression can complement *mir1Δ* phenotypes and mitochondria from *mir1Δpic2Δ* yeast strains regain phosphate transport activity when PIC2 is overexpressed (*Hamel et al., 2004*), suggesting that phosphate can be a PIC2 substrate. However, it is unlikely that this transport activity is physiologically relevant under normal conditions as *PIC2* deletion does not result in phosphate deficiency phenotypes in yeast. Based on these findings, we predict that while yeast PIC2 and MIR1 have specialized to transport specific substrates, PIC2 retains some promiscuity for both Cu and phosphate transport. In contrast, humans express a single paralog of PIC2/MIR1, SLC25A3, which serves as the major mitochondrial transporter of both Cu and phosphate (*Boulet et al., 2018*; *Kwong et al., 2014*). Cells lacking *SLC25A3* exhibit a Cu-dependent COX assembly defect (*Boulet et al., 2018*). Additionally, SLC25A3 transports Cu when recombinantly expressed and reconstituted in liposomes or when heterologously expressed in *L. lactis* (*Boulet et al., 2018*). Similarly, both phenotypic and biochemical assays confirm that SLC25A3 is the major phosphate transporter in mammalian mitochondria (*Kwong et al., 2014*; *Fiermonte et al., 1998*; *Wohlrab et al., 1986*).

These findings highlight a major unanswered question in our understanding of MCFs. Specifically, what differences enable the transport of single versus multiple substrates? Using newly available phylogenomic data from diverse lineages that span the major eukaryotic supergroups, we used an evolutionary framework to infer residues in the PIC2-MIR1 MCF subfamily that likely mediate substrate selection and transport. By coupling phylogenetic analyses with biochemical assays, we have uncovered residues required for transport of Cu and phosphate. Further, we demonstrate that Cu transport to the mitochondrial matrix is directly responsible for the COX deficiency observed in cells lacking *SLC25A3*.

## Results

### MIR1 does not transport Cu

To determine if MIR1 can transport Cu in addition to phosphate, we exploited the fact that MCF proteins insert into the cytoplasmic membrane of *L. lactis* in an active state and that Cu transport activity in this system can be detected by growth arrest in the presence of Ag$^+$ (*Figure 1A*; *Monné et al., 2005*; *Vest et al., 2013*). This assay was also used to assess phosphate transport by quantifying the growth rates of *L. lactis* strains expressing MCF genes in the presence of the toxic phosphate mimetic arsenate (AsO$_4^{3-}$). In the presence of 80 μM Ag$^+$, the growth of *L. lactis* expressing PIC2, but not MIR1 or an empty vector (EV), was significantly inhibited (*Figure 1B*). In contrast, the growth of *L. lactis* expressing MIR1 or PIC2 was inhibited to the same extent when cultured in

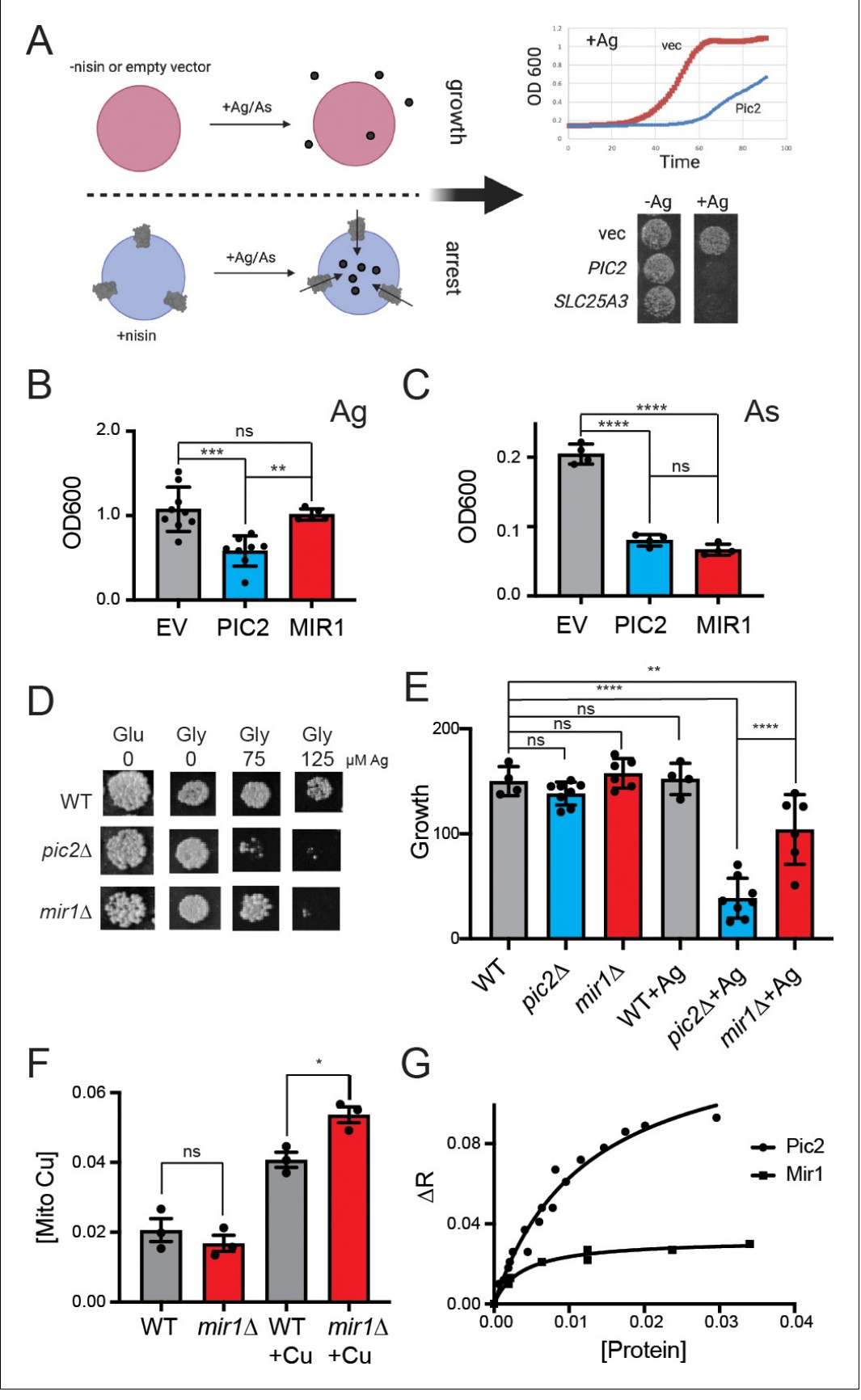

**Figure 1.** *S.cerevisiae* MIR1 does not transport Cu. (**A**) Schematic representation of the *L. lactis* expression system used to quantify transport characteristics. Survival is determined by the growth rate in liquid culture or by visual inspection of cells grown on agar plates containing $Ag^+$ or $AsO_4^{3-}$ in the presence of the inducer nisin. (**B**) Quantification of the growth of *L. lactis* expressing empty vector (EV), *S. cerevisiae* PIC2, or *S. cerevisiae* MIR1 after 12 hr in 80 µM $Ag^+$-containing media (n > 5). (**C**) Quantification of the growth of *L. lactis* expressing EV, PIC2, or MIR1 after 12 hr in 1.6 mM $AsO_4^{3-}$-containing media (n = 5). (**D**) Wild-type (WT), *pic2∆*, or *mir1∆* yeast grown in rich medium with a fermentable (Glu: glucose) or a non-fermentable (glycerol: Gly) carbon source in the absence (0) or presence of $Ag^+$ (75 or 125 µM). All strains were spotted on media as a $10^{-3}$ dilution of $OD_{600}$ of 1. (**E**) Densitometry measurements of serial dilutions (10, $10^2$, $10^3$, $10^4$) of cells in **D** on Glu, Gly, and Gly plus 75 µM Ag (WT n = 4, *pic2∆* n = 8, *mir1∆* n = 6). (**F**) Cu content of purified intact mitochondria from *mir1∆* cells assayed by Inductively coupled plasma - optical emission spectrometry (ICP-OES) and compared with that of parental WT cells. Both strains were grown in YP medium with glucose as a carbon source containing 10 µM bathocuproinedisulfonic acid (BCS) or 100 µM Cu (+Cu) (n = 3). (**G**) Fluorescence anisotropy (FA) of CuL (Ex320, Em400) upon the addition of reconstituted PIC2 or MIR1 in proteoliposomes prepared from extracted egg yolk lipids. Control FA of equal quantity of lipids without protein added was subtracted from each data point. Protein concentrations were determined by Bradford assay, and curves are fit with a nonlinear regression that assumes a single binding site. In all panels, data are plotted as the mean ± standard deviation and a one-way ANOVA was used for statistical analysis; ns: not statistically significant; *p<0.05, **p<0.01, ***p<0.001, ****p<0.0001.

1.6 mM $AsO_4^{3-}$ relative to a control strain harboring the EV (*Figure 1C*). These data show that in *L. lactis* MIR1 is capable of transporting the phosphate mimetic $AsO_4^{3-}$ but not the Cu mimetic $Ag^+$.

Consistent with our previous results (*Vest et al., 2013*), we find that the growth of yeast lacking *PIC2* is severely compromised on a non-fermentable carbon source in the presence of 75 µM $Ag^+$ due to a Cu deficiency in mitochondria (*Figure 1D, E*). In contrast, yeast lacking *MIR1* only exhibited a mild growth defect relative to the isogenic WT strain at this $Ag^+$ concentration (*Figure 1D, E*). Exposure to 125 µM $Ag^+$ led to a growth defect in both *mir1∆* and *pic2∆* yeast but not in the isogenic WT strain (*Figure 1D*). To further establish that MIR1 is incapable of Cu transport activity, we quantified mitochondrial Cu levels by inductively coupled optical emission spectroscopy. Cu levels in mitochondria from *mir1∆* yeast cells were similar to those isolated from WT cells (*Figure 1F*). In yeast mitochondria, Cu is stably bound by a fluorescent, non-proteinaceous ligand (CuL). Previously we used fluorescence anisotropy to investigate the binding of this fluorescent complex to purified PIC2 and SLC25A3 (*Boulet et al., 2018*; *Vest et al., 2016*; *Vest et al., 2013*). The decreased levels of anisotropy observed for purified MIR1 compared to PIC2 showed limited interaction with the CuL complex and MIR1 (*Figure 1G*). Thus, while the growth assays indicate that *MIR1* deletion can produce a Cu-dependent respiration defect at high $Ag^+$ concentrations, our biochemical data suggest that MIR1 does not transport Cu. Therefore, both MIR1 and PIC2 transport phosphate but only PIC2 can transport Cu.

## Mitochondrial Cu and phosphate carriers duplicated early in the evolution of eukaryotes

It is not surprising that MCF proteins are present across all eukaryotes given their fundamental roles in maintaining cellular physiology. We hypothesize that Cu transport to mitochondria was an important consideration in eukaryogenesis as it is required to maintain the activity of the electron transport chain and provide an advantage to the ancestral eukaryote (*Cobine et al., 2021*). Conservation of this activity across diverse organisms may provide a phylogenetic signal with which to resolve residues that dictate PIC2 and MIR1 substrate specificities. One hypothesis is that because ancient proteomes were smaller the transporters in these organisms were generalists that gained specificity as a consequence of gene duplication and subsequent subfunctionalization (*Risso et al., 2014*; *Risso et al., 2013*; *Eick et al., 2017*; *Eick et al., 2012*).

To provide evolutionary context for the existing experimental data, which has nearly all been collected from mammals and yeast, we performed a phylogenetic analysis on MCF transporters from a broad range of eukaryotic lineages. We selected a set of 47 taxa that spanned the supergroups within the eukaryotic Tree of Life (eToL) (*Burki et al., 2020*; *Supplementary file 1*, *Figure 2—figure supplement 1*). Only taxa with complete nuclear and mitochondrial genome sequences were included to accurately enumerate gene duplications and losses, and to ensure that apparent losses

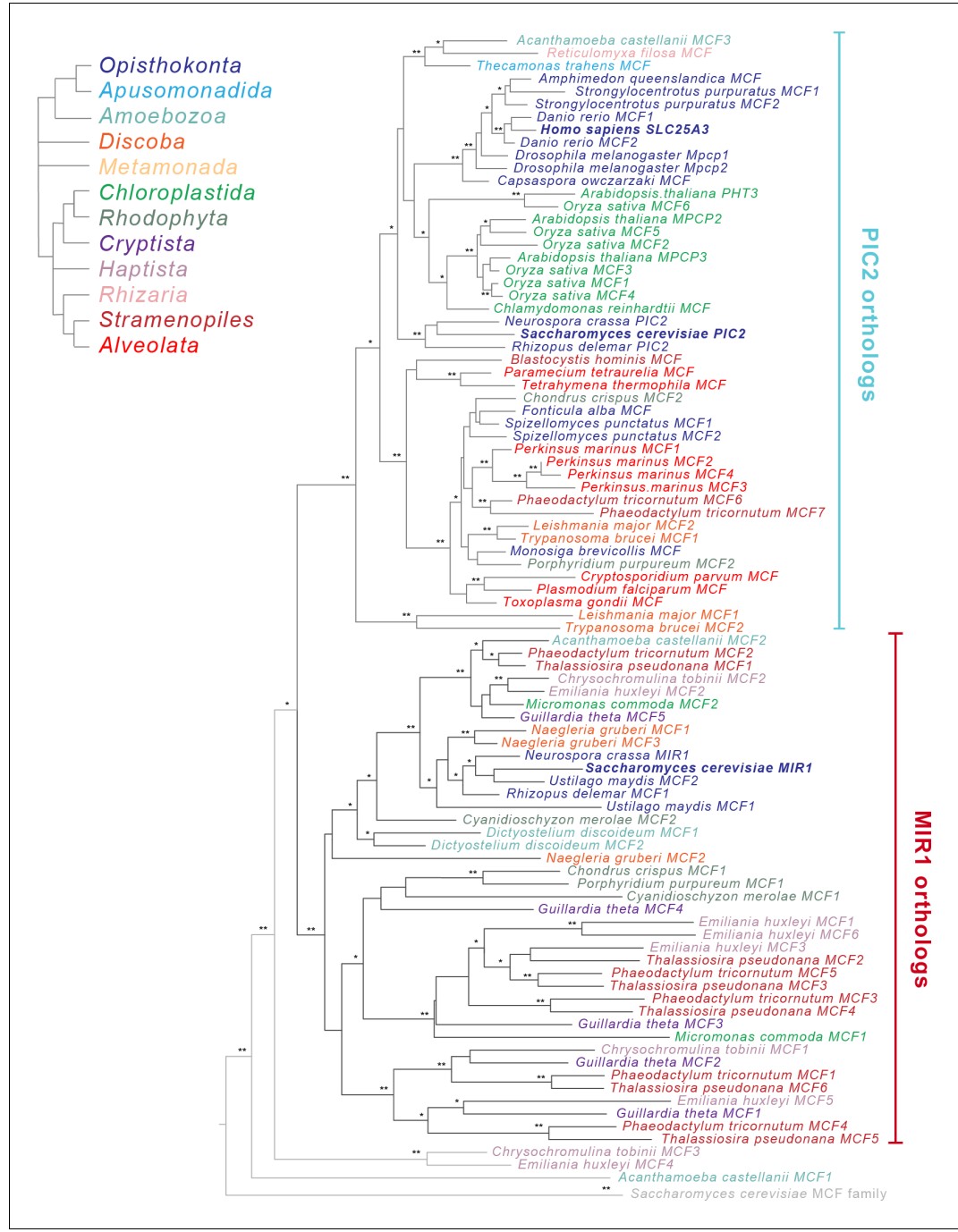

**Figure 2.** Phylogenetic analysis of the PIC2/MIR1 orthologs from 47 taxa reveals two major clades. Amino acid sequences of the eukaryotic MIR1/PIC2/SLC25A3 orthologs were aligned with the complete set of mitochondrial carrier family (MCF) proteins from *S. cerevisiae*. The maximum-likelihood tree shown was constructed in iQ-TREE using a general codon exchange matrix for nuclear genes with amino acid frequencies determined empirically from the data and seven rate categories (LG+F+R7). Support for the nodes was calculated using 1000 replications and is indicated as follows: **>95%; *>75%. Taxa names for the MIR1/PIC2/SLC25A3 sequences are color-coded according to the eukaryotic Tree of Life supergroups as indicated; the *S. cerevisiae* MCF outgroup sequences (gray) have been collapsed to a single branch. Accession numbers for each of the sequences are available in *Supplementary file 1* and *Figure 2—figure supplement 2*.

The online version of this article includes the following figure supplement(s) for figure 2:

**Figure supplement 1.** Pipeline for bioinformatic analysis.

**Figure supplement 2.** Phylogenetic analysis of the PIC2/MIR1 orthologs.

*Figure 2 continued on next page*

*Figure 2 continued*
**Figure supplement 3.** Neighbor joining tree of mitochondrial carrier families (MCFs) from *Acanthamoeba castellanii.*

were not due to incomplete datasets. From these genomes, a total of 2447 putative MCF family members were identified based on the presence of a mitochondrial carrier domain (PFAM domain PF00153) (*Supplementary file 1*). To distinguish PIC2-MIR1 orthologs from other members of the MCF family, phylogenetic trees were constructed using the MCF proteins from each taxon as well as the complete set of yeast and human MCF proteins. Candidate sequences that clustered with PIC2 or MIR1 were retained for further analyses (92 of 2447 MCF sequences) (*Figure 2—figure supplement 1*).

The amino acid sequences of these potential Cu and/or phosphate transporting proteins were aligned and subsequently used to reconstruct the evolutionary history of PIC2-MIR1 orthologs across eukaryotes (*Figure 2*). Of the 92 sequences, 47 clustered with *S. cerevisiae* PIC2 and are referred to as PIC2-like while 42 clustered with *S. cerevisiae* MIR1 and are defined as MIR1-like. The remaining three sequences were more closely related to PIC2-MIR1 than other MCFs but nonetheless fell outside of these two well-supported clades (*Figure 2* and *Figure 2—figure supplement 3*).

To estimate the timing of gene duplications and losses within eukaryotes, we overlaid the presence and/or absence of PIC2-like or MIR1-like sequences onto the established eToL tree (*Figure 3A*). Recent phylogenomic analyses indicate that extant eukaryotes form nine supergroups (*Burki et al., 2020*). Species from seven of these groups were included in this analysis: Amorphea, Discoba, Archaeplastida, TSAR (Telonemids, Stramenopiles, Alveolates, and Rhizaria), Haptista, Cryptista, and Metamonada. Two additional groups, CRuMs (Collodictyonids, Rigifilida, and Mantamonas) and Hemimastigophora, were not included due to the lack of complete nuclear genome sequences. PIC2-MIR1 orthologs were present in each taxon analyzed with the exception of those from Metamonada, which are anaerobic protists that secondarily lost mitochondria (*Karnkowska et al., 2019*; *Karnkowska et al., 2016*). This broad phylogenetic conservation suggests that the two paralogs were present within the last common eukaryotic ancestor (*Figure 3A*).

Given the ancient origin of PIC2 and MIR1, we first analyzed the presence and absence of orthologs within Amorphea, which consists of the opisthokonts (animals, fungi, and yeast), apusomonads, and amoebae (*Burki et al., 2020*). MIR1-like sequences are absent from Holozoan taxa with this lineage retaining only PIC2-like transporters (*Figure 3A, B*). In contrast, the fungal lineages (Holomycota) exhibit more variability in the numbers of PIC2-like and MIR1-like sequences (*Figure 3B*). Single orthologs of each type are present in *S. cerevisiae* and the closely related *Neurospora crassa*. The only Amorphea taxa that lost PIC2 are *Ustilago maydis* and *Dictyostelium discoideum,* which both have a *MIR1* duplication. Outside the Amorphea, the gene copy number of the PIC2-MIR1 orthologs is more variable, which may reflect different evolutionary pressures on these transporters across lineages. Several lineages have lost either PIC2 or MIR1 and retained multiple copies of the remaining paralog (e.g., PIC2-like transporters within Chloroplastida and the alveolate *Perkinsus marinus* or the MIR1 duplications in Cryptista and Stramenopile lineages; *Figure 3*). This raises the possibility that, to compensate for the loss of the MIR1 transporter, PIC2 duplicated and convergently evolved additional substrate specificities. While there may be other constraints on this evolution, the loss of a PIC2 ortholog is always accompanied by duplication of the MIR1 ortholog. In contrast, a PIC2-like MCF is retained in all species that have a single PIC2-MIR1 ortholog, indicating that the loss of MIR1 does not always coincide with PIC2 duplication.

## Structural modeling of PIC2 suggests appropriate spatial organization of conserved residues that may coordinate Cu transport

We hypothesize that specific residues in PIC2-like proteins that confer the ability to transport Cu are absent in MIR1-like proteins, while amino acids conserved across PIC2- and MIR1-like proteins are required for both Cu and phosphate transport. To predict residues involved in substrate specificity, we modeled the PIC2 sequence onto the c-state and m-state structures of the ADP/ATP carrier (*Ruprecht et al., 2019*; *Pebay-Peyroula et al., 2003*; *Figure 4—figure supplement 1*). To quantify sequence conservation at the level of individual residues independently of the evolutionary histories

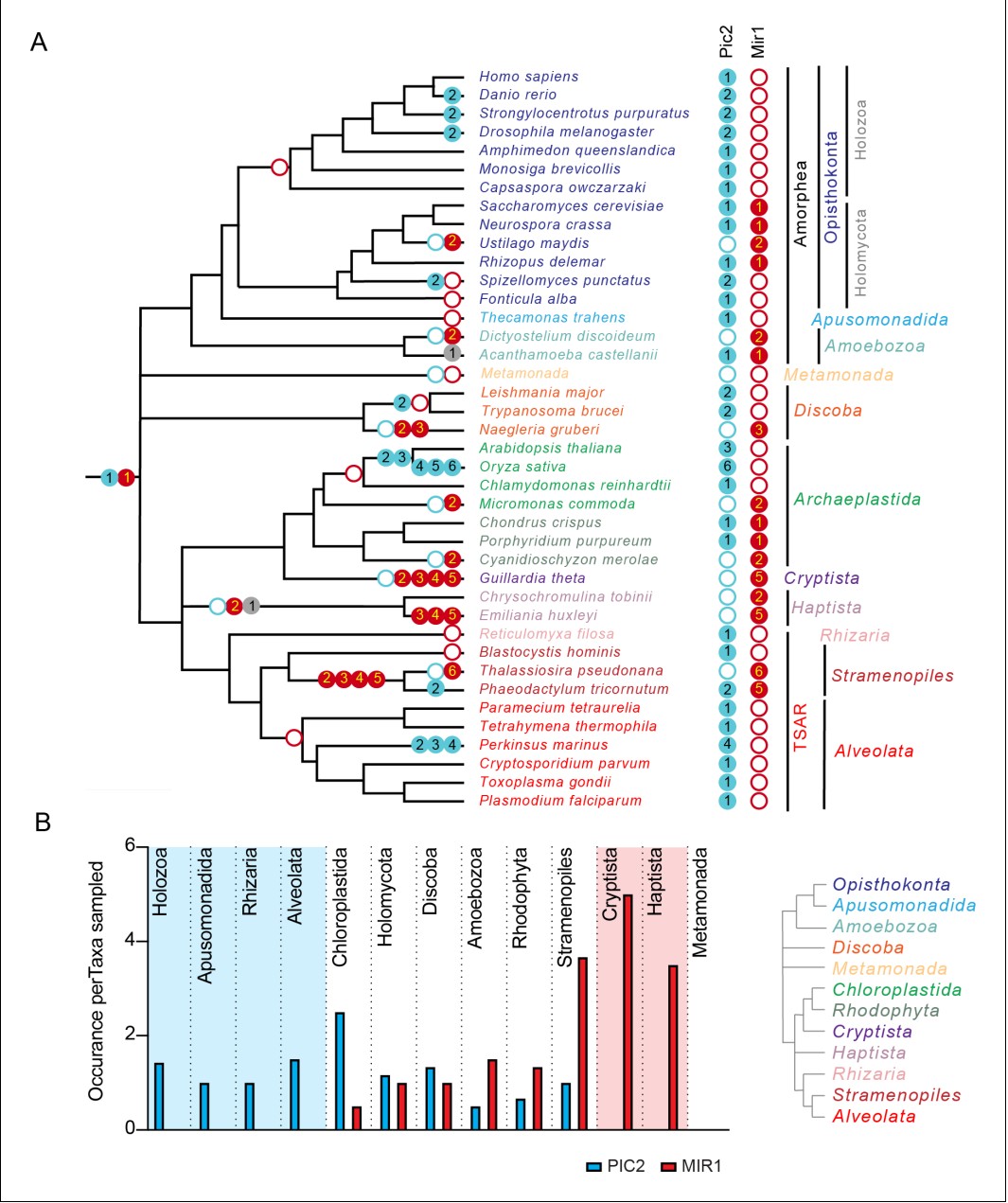

**Figure 3.** The PIC2/MIR1 family of mitochondrial carrier family transporters is ancient within eukaryotes. (**A**) Using the presence or absence of orthologs within the eukaryotic lineages, we inferred the evolutionary timings of gene duplications (solid circles) and losses (hollow circles) of the PIC2-like (blue), MIR1-like (red), and other (gray) sequences. (**B**) The average number of PIC2 and MIR1 orthologs identified in the sampled taxa from eight of the nine eukaryotic supergroups.

of the proteins, Shannon entropy was calculated for each position within an alignment of the PIC2-like sequences (**Figure 4A, B**; **Supplementary file 1**). Shannon entropy is one of the simplest and most common measures of conservation that can be calculated from multiple sequence alignments (**Capra and Singh, 2007**). The Shannon entropy was calculated for each residue within alignments of the PIC2-like sequences and compared to the values determined from the complete PIC2-MIR1 grouping (**Supplementary file 1**). By integrating the structural models and phylogenetic analyses, we were able to visualize conserved residues as a surface representation (**Figure 4C–E**). The PIC2-like orthologs show high conservation (Shannon entropy <0.5 suggesting that the residue is

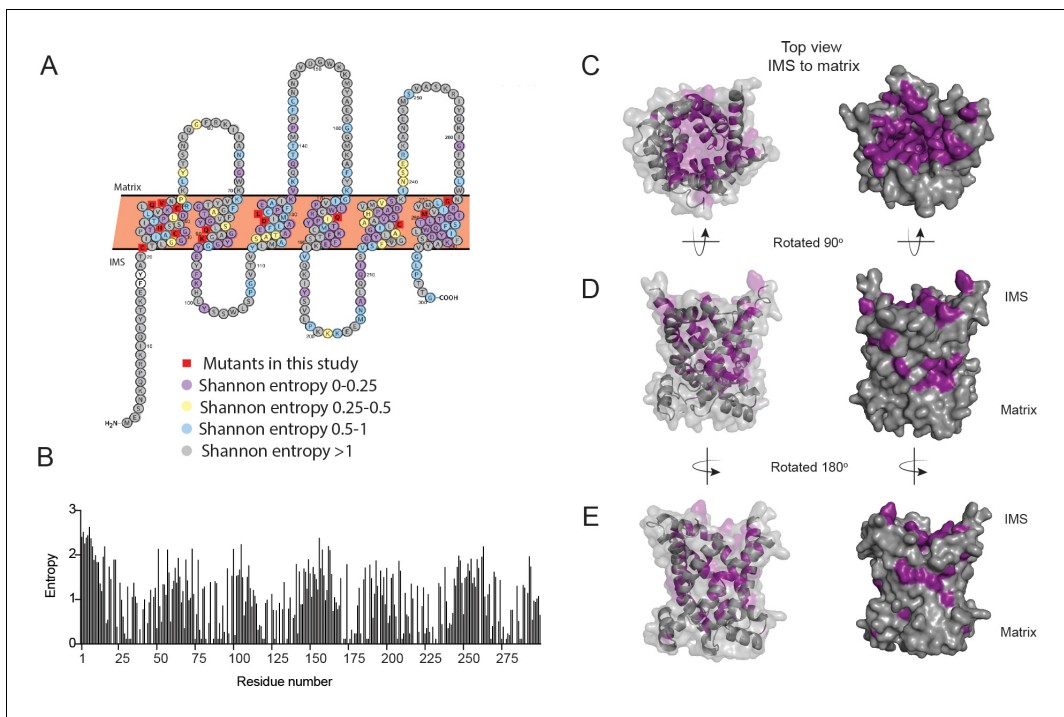

**Figure 4.** Conservation of residues in PIC2. (**A**) A Protter representation of the PIC2 amino acid sequence was generated and colored based on Shannon entropy scores for conservation of a given residue. (**B**) The Shannon entropy for each residue in PIC2 based on all sequences in the PIC2-specific clade (see ***Supplementary file 1***). (**C**) Structure of PIC2 in the c-state viewed from the intermembrane space side, with conserved residues (Shannon entropy <0.5) highlighted in purple and all other residues shown in gray. (**D**) A 90° rotation of the structure to view it from side, and (**E**) a 180° rotation to view it from the opposite side.

The online version of this article includes the following figure supplement(s) for figure 4:

**Figure supplement 1.** Structural models of PIC2.

**Figure supplement 2.** Conservation surface of PIC2 viewed from the intermembrane space.

maintained across all forms of the protein in the multiple sequence alignment) in the aqueous binding pocket, while alignment with the complete PIC2-MIR1 family further reveals a smaller subset of conserved residues (***Figure 4—figure supplement 2***). This analysis also detects conserved patches extending into the IMS and outside the aqueous binding pocket in the lipid bilayer that may be required for interactions with other components of the IM (***Figure 4D, E***).

To identify residues that mediate Cu transport, we initially focused on the well-established Cu-binding ligands Cys, His, and Met. Analysis of the PIC2-MIR1 ortholog trees showed that histidine 33 (all residues are numbered according to the yeast PIC2 sequence) is conserved in both the PIC2 and MIR1 clades (***Figure 5***). Cysteine 29 is conserved in the PIC2 clade and most MIR1 proteins but is replaced with alanine in the MIR1-like transporters from lineages with multiple duplications (*Emiliania huxleyi, Thalassiosira pseudonana,* and *Phaeodactylum tricornutum*) (***Figure 5***, ***Figure 5—figure supplement 1***). Cysteine 21 and Cys225 are strictly conserved among PIC2 orthologs, but not among MIR1 orthologs (***Figure 5***). Cysteine 44 is conserved in the PIC2-like clade, while MIR1-like orthologs have a conserved threonine in the equivalent position (***Figure 5***). The PIC2-like transporters that lack Cys44 are the *P. marinus* duplications, one of two copies of PIC2 in *P. tricornutum*, and the single copy of PIC2 in *N. crassa*. Analysis of the structural models revealed that Cys21, Cys29, Cys44, and His33 are positioned along one side of the aqueous binding pocket (***Figure 5—figure supplement 2***), whereas Cys225 is on the opposite side of this pocket. Cysteine 225 is positioned to interact with the peptide backbone of Cys182 (based on the alignments, this residue is only a cysteine in *S. cerevisiae*), which faces away from the aqueous binding pocket. Together, these data suggest that Cys21, Cys29, Cys44, and His33 may combine to form transient sites that bind Cu directly as it moves through the IM.

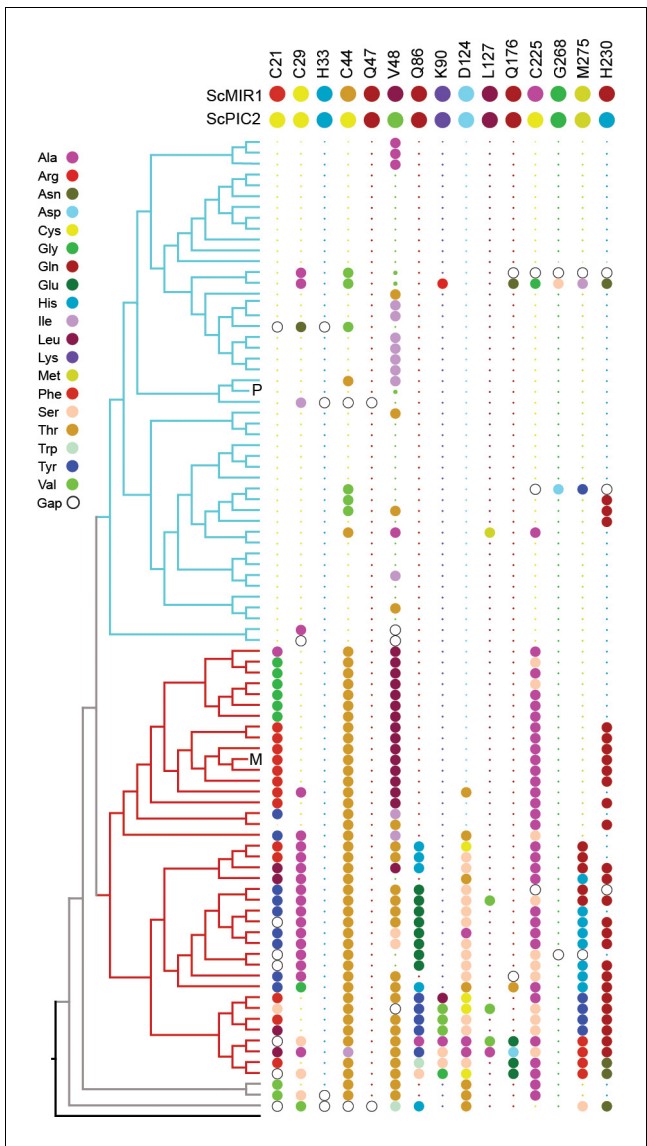

**Figure 5.** Conservation of selected residues in the PIC2/MIR1 family of transporters. The tree topology is identical to that shown in *Figure 2*. Amino acids are colored according to the key, and insertion/deletion events that lead to gaps within the alignment are indicated by the hollow circles. P indicates position of *S. cerevisiae* PIC2, and M indicates *S. cerevisiae* MIR1. Small dots indicate that the residue is identical to that of PIC2 (shown at the top), and large dots indicate differences.

The online version of this article includes the following figure supplement(s) for figure 5:

**Figure supplement 1.** Map of the residues found in gene duplications.

**Figure supplement 2.** Position of residues mutated in the aqueous binding pocket of PIC2.

## Mutating structural elements and conserved contact points cause differential transport defects

To assess the functional importance of the Cys-His residues in Cu and/or phosphate transport, we altered these residues in the context of *S. cerevisiae* PIC2 and expressed the mutants in *L. lactis*. To verify expression, protein content was assessed in *L. lactis* cells expressing the mutants versus EV controls. Although the levels of heterologous expression were too low to observe using Sypro Ruby (*Figure 6—figure supplement 1*), the mutant proteins were readily detectable upon immunoblot analysis (*Figure 6A*, *Figure 6—figure supplement 2*). To assay Cu transport, we cultured each

variant in media containing an $Ag^+$ concentration that inhibited growth of *L. lactis* expressing WT PIC2 but not of cells harboring an EV (*Figure 1A*, *Figure 6*). *L. lactis* expressing C21A, C29A, H33A, C44A, and C225A PIC2 mutants showed equal expression levels to WT PIC2 (*Figure 6A*) but displayed $Ag^+$ resistance relative to *L. lactis* expressing WT PIC2 (all p<0.012) (*Figure 6B*), with the most resistance observed in the H33A mutant. However, these mutants also exhibited a growth defect relative to cells with an EV, suggesting that residual transport activity is present. Similarly, when $Ag^+$ was replaced with $AsO_4^{3-}$ to assess phosphate transport, *L. lactis* expressing each of the five PIC2 mutants displayed increased resistance to $AsO_4^{3-}$ (*Figure 6C*), suggesting that these mutations also limit its transport.

Computational analyses predict that *S. cerevisiae* MCF transporters have three contact sites for substrate binding (*Robinson et al., 2008*). In PIC2, the proposed phosphate substrate contact points are Gln86 and Lys90 in transmembrane helix (TMH) 2, Gln176 in TMH4, and Met275 in TMH6 (*Figure 4*; *Ruprecht and Kunji, 2020*; *Ruprecht et al., 2019*; *Robinson et al., 2008*). These residues are largely conserved in both the PIC2-like and MIR1-like clades (*Figure 5*), as is expected for transporters that share a substrate. We mutated each of these residues to alanine and assessed transport activity as described above. When expressed in *L. lactis,* the Q86A and Q176A mutants were expressed at WT levels (*Figure 6A*) and were more resistant to $Ag^+$ than WT PIC2 (*Figure 6B*) but less resistant than cells expressing EV. In contrast, the K90A and M275A mutants exhibited comparable $Ag^+$ sensitivity to WT PIC2 (p>0.05), suggesting that these substitutions do not affect Cu transport (*Figure 6B*). The addition of $AsO_4^{3-}$ to the media only inhibited the growth of cells expressing WT PIC2; cells expressing Q86A, K90A, Q176A, and M275A all grew at similar rates as cells expressing the EV (*Figure 6C*).

Finally, we interrogated the functional significance of a subset of residues that were selected based on sequence conservation and our structural model: Gln47, Val48, Asp124, Leu127, and Gly268 (*Figure 4A*, *Figure 5—figure supplement 2*). With very few exceptions, Gln47 is conserved among eukaryotic PIC2-MIR1 orthologs (*Figure 4A, B* and *Figure 5*). Val48 is part of a group of residues that appear to close the aqueous binding pocket in the c-state (*Figure 5—figure supplement 2*). Asp124 interacts with Gln176 (*Figure 5—figure supplement 2*) and is conserved among all PIC2-like orthologs and those transporters most closely related to yeast MIR1 (*Figure 5*). Leu127 is conserved in all orthologs and interacts with Gln86 (*Figure 4A, B*, *Figure 5*, *Figure 5—figure supplement 2*). Gly268 is maintained throughout the evolution of this protein family (*Figure 4A*, *Figure 5*). The Q47A variant was unstable in *L. lactis* (*Figure 6A*), suggesting that it has been maintained across evolution for structural stability. The V48A variant did not affect protein expression (*Figure 6—figure supplement 2*) or $AsO_4^{3-}$ resistance. However, it did result in a significant difference in $Ag^+$ resistance compared to WT PIC2 (*Figure 6B, C*). The D124A PIC2 mutant was stably expressed in *L. lactis* (*Figure 6A*) and more resistant to $Ag^+$ than WT PIC2 (*Figure 6A*) but less resistant than cells expressing EV, suggesting that it harbored residual Cu transport activity. When expressed in *L. lactis,* the L127A PIC2 variant showed WT expression (*Figure 6A*) and equivalent susceptibility to $Ag^+$ as WT PIC2 but was resistant to $AsO_4^{3-}$ (*Figure 6B, C*), indicating that this single substitution interferes with phosphate transport but does not prevent Cu transport. Finally, the G268A variant was unstable in *L. lactis* (*Figure 6A*, *Figure 6—figure supplement 2*), suggesting that the increased resistance to $Ag^+$ and $AsO_4^{3-}$ associated with the expression of this variant was due to decreased levels of the protein (*Figure 6B, C*). We also tested a series of mutants that exchanged the residues found in yeast PIC2 and mammalian SLC25A3 with those found in MIR1. Conversion of the PIC2 residues Ser102, Tyr156, Thr180, Gln138, Glu242, and Val191 to the equivalent residues in MIR1 did not affect the ability to transport $Ag^+$ (*Figure 6—figure supplement 3*). Collectively, the data from the *L. lactis* assays show that mutating individual residues can impair the transport of both substrates, or Cu or phosphate alone.

## Mitochondrial transport of phosphate but not Cu is compromised in a Leu175 mutant of SLC25A3

Based on the His33 and Leu127 PIC2 mutant data from *L. lactis*, we investigated the transport activity of the equivalent variants in murine SLC25A3 (His75 and Leu175). Consistent with the failure of the H33A PIC2 mutant to transport $Ag^+$ or $AsO_4^{3-}$ in *L. lactis*, expression of the H75A SLC25A3 variant in immortalized mouse embryonic fibroblasts (MEFs) with floxed (WT) or collapsed (KO) *Slc25a3* alleles did not rescue the COX deficiency of the knock-out (KO) cells (*Figure 7A, B*). Conversely,

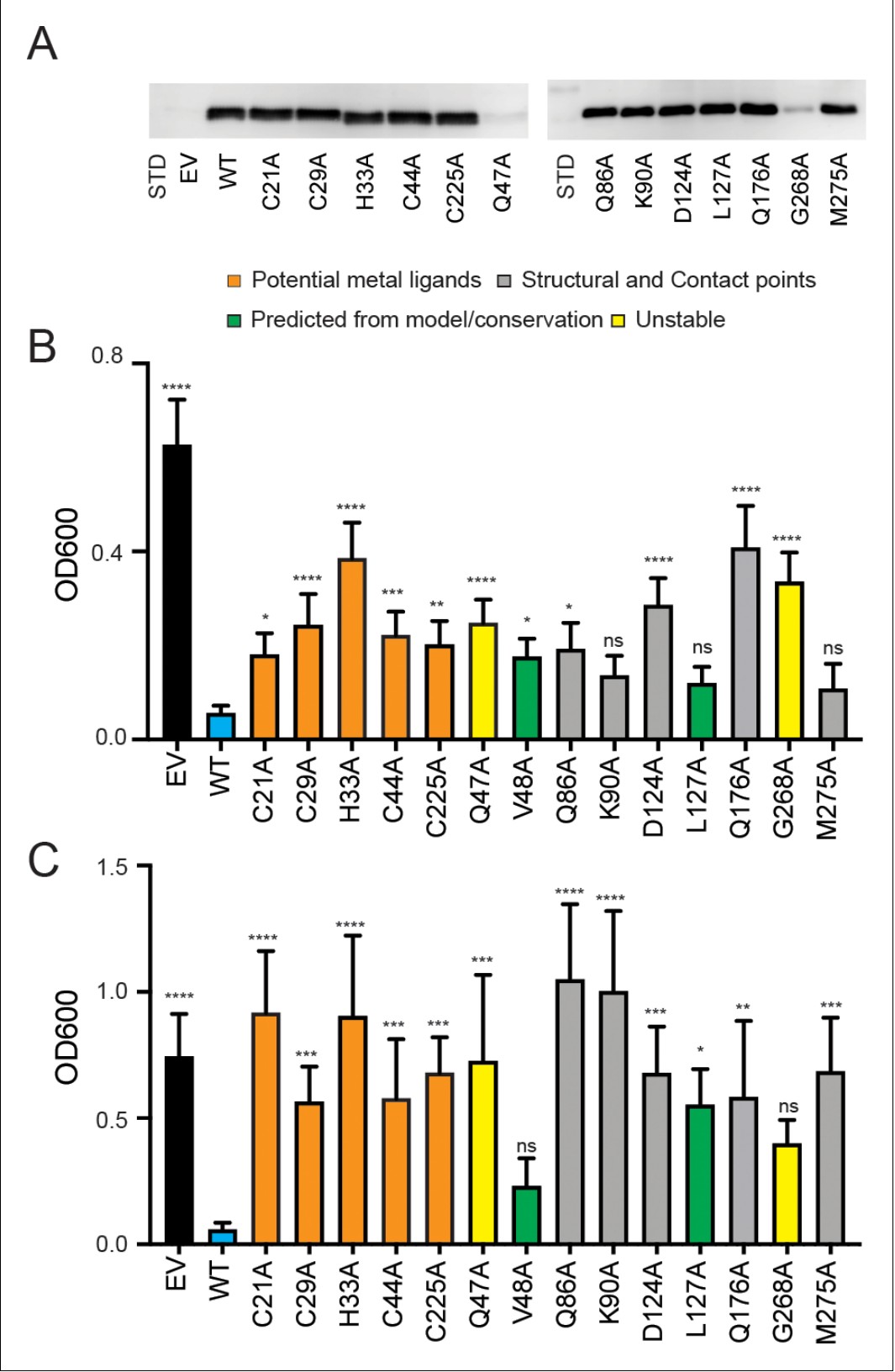

**Figure 6.** Expression of PIC2 and variants in *L. lactis.* (**A**) Immunoblot of *L. lactis* extracts expressing empty vector (EV), wild-type PIC2 (WT), or a given PIC2 variant in which each of the listed residues was converted to an alanine prepared from equal numbers of cells based on optical density after induction with nisin. (**B**) Growth of *L. Figure 6 continued on next page*

*Figure 6 continued*

*lactis* expressing EV, wild-type PIC2 (WT), or a given PIC2 variant in Ag⁺-containing media. Each bar represents the median of 10–18 independent cultures with 95% confidence interval as error bars (*p<0.05, **p<0.01, ***p<0.001, ****p<0.0001 based on one-way ANOVA relative to PIC2 wild-type control). The color of the bar indicates one of four major groupings: Cu-binding (orange), structural motifs or contact points (gray), evolutionarily conserved and present in the aqueous binding pocket of the transporter (green), and unstable in *L. lactis* (yellow). (C) As described in (B) except *L. lactis* strains were grown in $AsO_4^{3-}$-containing media.

The online version of this article includes the following figure supplement(s) for figure 6:

**Figure supplement 1.** Protein expression in *L. lactis*.
**Figure supplement 2.** Immunoblot analysis of PIC2 expressed in *L. lactis*.
**Figure supplement 3.** Substitution of PIC2 residues for MIR1 residues.

expression of the L175A SLC25A3 variant was able to reverse the COX defect (*Figure 7B*). Immunoblot analysis showed that the L175A mutant was present in mitochondria and increased steady-state COX1 levels (*Figure 7C*). Consistent with our previous studies using a mitochondrially targeted Cu sensor (*Dodani et al., 2011*; *Boulet et al., 2018*), we found that total mitochondrial Cu content was significantly reduced in KO MEFs and increased in KO MEFs expressing the L175A variant (*Figure 7D*).

Reconstitution of MCF proteins in liposomes has been used extensively to assess substrate transport and specificity (*Marobbio et al., 2015*; *Marobbio et al., 2003*; *Fiermonte et al., 1998*; *Fiermonte et al., 2003*; *Cavero et al., 2003*; *Catalina-Rodriguez et al., 2012*). Liposomes created from mitochondrial membranes of WT but not KO MEFs were able to transport Cu (*Figure 7E*). The Cu transport defect in KO-derived liposomes was reversed upon expression of the L175A variant (*Figure 7E*). To assess phosphate uptake, mitochondrial swelling in the presence of phosphate was measured (*Kwong et al., 2014*; *Hamel et al., 2004*). Intact mitochondria isolated from KO cells had a phosphate uptake defect compared to WT that was rescued by expressing WT SLC25A3 but not the L175A variant (*Figure 7F*). Taken together, these data show that the L175A mutant is able to transport Cu but not phosphate in mitochondria and that this Cu transport activity is sufficient to rescue COX activity.

## Discussion

The mechanisms that mediate MCF transporter specificity remain largely unknown. Although two recent studies have shown that single residue changes can modulate MCF substrate specificity (*Knight et al., 2019*; *King et al., 2020*), the majority of investigations have focused on deficiencies in the transport of one substrate, and few have assessed substrate promiscuity. Here, we directly addressed this issue by focusing on Cu and phosphate transport, which, in mammals, is mediated by the single MCF transporter SLC25A3. Multiple studies clearly connect SLC25A3 to phosphate transport, and mutations in *SLC25A3* lead to skeletal muscle myopathy and heart disease in humans (*Boulet et al., 2018*; *Seifert et al., 2016*; *Bhoj et al., 2015*; *Kwong et al., 2014*; *Mayr et al., 2011*; *Mayr et al., 2007*) and cardiac hypertrophy in mice (*Kwong et al., 2014*). MEFs derived from the heart-specific *Slc25a3* knockout mouse exhibit clear COX and SOD1 defects that can be rescued by overexpression of a *Slc25a3* cDNA or addition of Cu (*Boulet et al., 2018*). These data are complemented by in vitro assays in liposomes showing Cu transport by purified SLC25A3 and by Ag⁺ growth arrest phenotypes associated with its expression in *L. lactis* (*Boulet et al., 2018*). The data presented in this study provide the first experimental evidence of a missense mutation that separates Cu and phosphate transport, and firmly establish that physiological defects in COX and SOD1 are due to impaired Cu transport rather than a secondary consequence of decreased phosphate transport.

### Evolutionary history of mitochondrial Cu–phosphate transporters

Our evolutionary analyses of the Cu–phosphate transporters were prompted by the observation that *S. cerevisiae* PIC2 and MIR1 exhibit substrate specificity, whereas the mammalian ortholog SLC25A3 is responsible for the transport of both Cu and phosphate. Selection on genes with multiple functions can constrain diversity to avoid negative effects associated with losing one of these functions.

Therefore, gene duplications serve as important sources for evolutionary selection and refinement. Resulting duplications can be retained for the original function, specialized for new functions, refined to enhance an existing function or allow for increased expression by gene dosage; if none of these outcomes occur, the duplicate gene is lost (*Kuang et al., 2016*; *Conant et al., 2014*; *Sandegren and Andersson, 2009*; *Conant and Wolfe, 2007*; *Hittinger and Carroll, 2007*; *Zhang et al., 2002*; *Force et al., 1999*). In *S. cerevisiae*, PIC2 and MIR1 are partially redundant for phosphate transport (*Hamel et al., 2004*). However, mutation of *MIR1* in *S. cerevisiae* is sufficient to produce phosphate-related phenotypes, suggesting that, under most conditions, the ability of PIC2 to transport phosphate is unable to compensate for loss of MIR1 function (*Boulet et al., 2018*; *Hamel et al., 2004*). Instead, the *PIC2* sequence appears to be optimized for Cu transport. Similarly, we show here that MIR1 lacks clear Cu transport activity even though *mir1Δ* yeast exhibit increased susceptibility to Cu restriction compared to WT cells. Our phylogenetic analyses of *PIC2* and *MIR1* sequences suggest that the gene duplication that created these two orthologs was an ancient event, and that evolutionary interplay between these two substrate specificities may have occurred multiple times throughout eukaryotic evolution.

The loss of *MIR1* has occurred multiple times in eukaryotes, an event that is likely facilitated by the dual specificity of PIC2. *SLC25A3* is essential in mammals as the homozygous deletion is embryonic lethal. While mammals do express two SLC25A3 isoforms, isoform A is expressed primarily in heart and skeletal muscle whereas isoform B is expressed in all tissues (*Fiermonte et al., 1998*; *Seifert et al., 2016*; *Kwong et al., 2014*). Therefore, it is unlikely that the isoforms provide the functional redundancy that would be afforded via gene duplication or retention of *MIR1*.

## Understanding Cu transport

Copper transport in eukaryotic cells has been an area of intense research since the discovery of cytosolic copper chaperones (*Robinson and Winge, 2010*; *Pufahl et al., 1997*), and the observation that there is vanishingly little freely available copper in the cytosol (*Rae et al., 1999*). These early findings

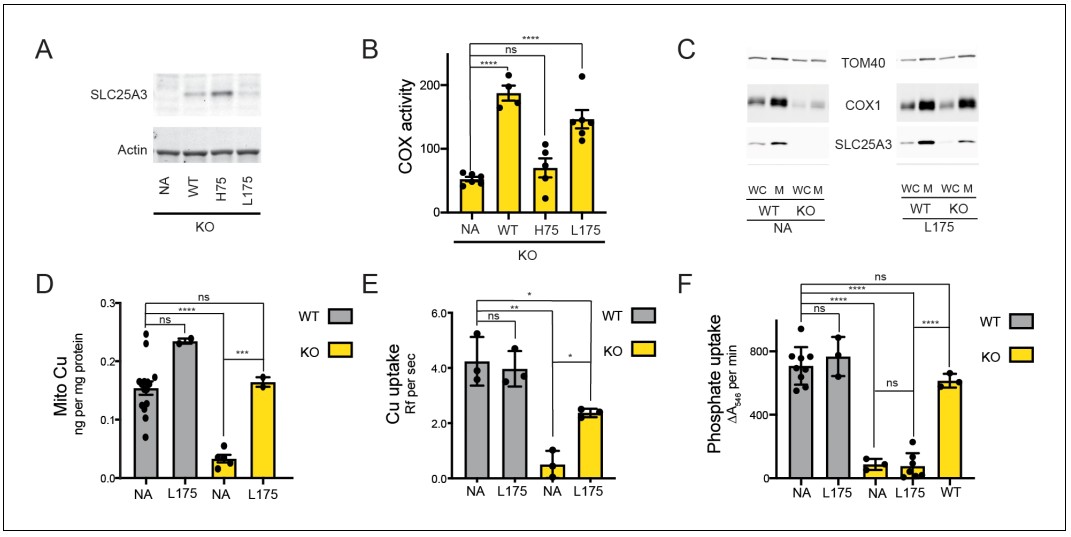

**Figure 7.** The SLC25A3 L175A variant restores mitochondrial Cu levels and rescues the cytochrome *c* oxidase (COX) deficiency in KO MEFs. (**A**) Immunoblot analysis of SLC25A3 abundance in *Slc25a3 KO* MEFs alone or those transduced with wild-type SLC25A3 (WT), a H75A variant (H75), or a L175A variant (L175). Actin served as an internal loading control. (**B**) COX activity in KO MEFs alone (n = 6) or transduced with WT SLC25A3 (n = 4), a H75A variant (H75) (n = 5), or a L175A variant (L175) (n = 6). ns, p>0.05, ****, p<0.0001 based on a one-way ANOVA. (**C**) Immunoblot analysis of SLC25A3, TOM40, and COX1 abundance in whole cells (WC) or isolated mitochondrial (M) from WT or KO MEFs alone (NA) or transduced with the SLC25A3 L175A variant (L175). (**D**) Total Cu levels in mitochondria from WT or KO cells as in (**C**), determined by ICP-OES. (**E**) Cu uptake in mitochondrially derived liposomes created by the membranes of mitochondria in (**C**) with additional lipids. Liposomes contain Phen Green to monitor the uptake of Cu. (**F**) Mitochondrial swelling rate in the presence of phosphate as a measure of phosphate uptake.

have been refined to recognize that, in addition to proteins, multiple cytosolic ligands contribute to the regulation of metal trafficking and target binding (*Waldron et al., 2009*). The recruitment of Cu to mitochondria was initially attributed to COX17 due to its dual localization in cytosol and IMS (*Glerum et al., 1996*). However, COX is fully functional when COX17 is artificially restricted to the IMS by an IM tether (*Maxfield et al., 2004*), suggesting that its critical role in holoenzyme assembly involves local, redox-regulated delivery of Cu to the accessory proteins SCO1/2 and COX11 (*Banci et al., 2008*; *Horng et al., 2004*). Consistent with a mitochondrially restricted function for COX17 in Cu handling, yeast cells lacking this gene accumulate wild-type levels of Cu in mitochondria (*Cobine et al., 2004*). In fact, attempts to isolate a protein that delivers Cu to mitochondria led to the identification of a non-proteinaceous ligand (CuL) that accumulates in the matrix (*Cobine et al., 2021*; *Vest et al., 2019*; *Vest and Cobine, 2011*; *Cobine et al., 2006*; *Cobine et al., 2004*). Although the molecular identity of this ligand remains unknown, its biophysical properties have been used to suggest that the ligand contributes to buffering cytosolic Cu and facilitating uptake of Cu into mitochondria (*Cobine et al., 2006*; *Cobine et al., 2004*). PIC2 is able to transport both the CuL purified from the mitochondrial matrix as well as ionic Cu in both liposomes and the *L. lactis* system (*Vest et al., 2013*). It is unclear if the transport of the CuL proceeds as an intact complex or if Cu is released from the ligand during transport. The ionic Cu in our transport assays is $Cu^+$ due to the presence of an exogenous reductant (e.g., ascorbate) or the endogenous reductant menaquinone in *L. lactis* (*Abicht et al., 2013*), and there is no experimental evidence for other metal ions being transported by PIC2. The transport of ionic Cu could be a mechanism to limit cytosolic accumulation of Cu during Cu-overload-induced stress (*Vest et al., 2013*; *Cobine et al., 2021*). Crosslinking and damage of mitochondrial membranes induced by Cu has been observed in models of Cu overload, such as the Long–Evans Cinnamon rat (*Zischka et al., 2011*).

In SLC25A3, the L175A mutation separates Cu and phosphate transport by fully restoring COX activity and mitochondrial Cu levels without rescuing phosphate transport. This finding confirms that the COX defect in mutant cells is due to defective Cu transport rather than reduced phosphate levels. Further, our data suggest that compromising the phosphate transport function of PIC2 is easier than inactivating its Cu transport function. Mutations in a series of cysteine and histidine residues lining the aqueous binding pocket of the c-state model decrease, but do not eliminate, Cu transport. The PIC2 structural model indicates that the Cys29 and His33 would be the most likely location to form a Cu-binding site. The cysteine positioned above that site (residue 21) may help recruit Cu from the IMS and present it to Cys29-His33. In the m-state model, the Cys29-His33 proximity is maintained and the next potential ligand, Cys44, is exposed, allowing for potential relocation of the Cu.

To understand the transport of the CuL complex, we considered the net negative charge of the complex, which suggests that positively charged or hydrogen-bond donor residues within the aqueous binding pocket may stabilize this interaction, including those that participate in phosphate transport (Gln86 and Lys90) (*Cobine et al., 2004*). Though mutating Lys90 does not affect ionic $Ag^+$ transport in *L. lactis*, we nonetheless envision that this residue may be important to transport competency in vivo where the CuL is more abundant. Interestingly, in the m-state model of PIC2 the aromatic ring of the side chain of Tyr83 is located between the Cys29 and His33, raising the possibility that these residues could be used as direct ligands for ionic Cu transport and as a site for binding for the CuL through π-interactions with aromatic components of the ligand. NMR analysis of the CuL by proton and carbon spectrums shows the presence of an aromatic ring structure with proton chemical shifts of 6.5–8 ppm and carbon chemical shifts of 110–175 ppm (*Figure 8—figure supplement 1*). The ring structure is consistent with the fluorescent properties of the CuL complex (*Vest et al., 2013*; *Cobine et al., 2006*). The positioning of an aromatic ring between the Cys29-His33 site could mimic a hypothetical CuL-bound state (from the c-state), and the movement of the Tyr83 side chain into this position during substrate transport could facilitate the release of the complex from the Cys29-His33 site toward the matrix (*Figure 8—figure supplement 1*). While we cannot differentiate between possible mechanisms of transport that include release of Cu to the matrix upon CuL binding or direct transport of the intact CuL complex, the transport of intact CuL may be expected as this is the major form of Cu found in mitochondria under normal conditions (*Dodani et al., 2011*; *Cobine et al., 2006*; *Cobine et al., 2004*). In addition, the anionic nature of the CuL complex may explain some of the promiscuity between Cu and phosphate as substrates of the same carrier.

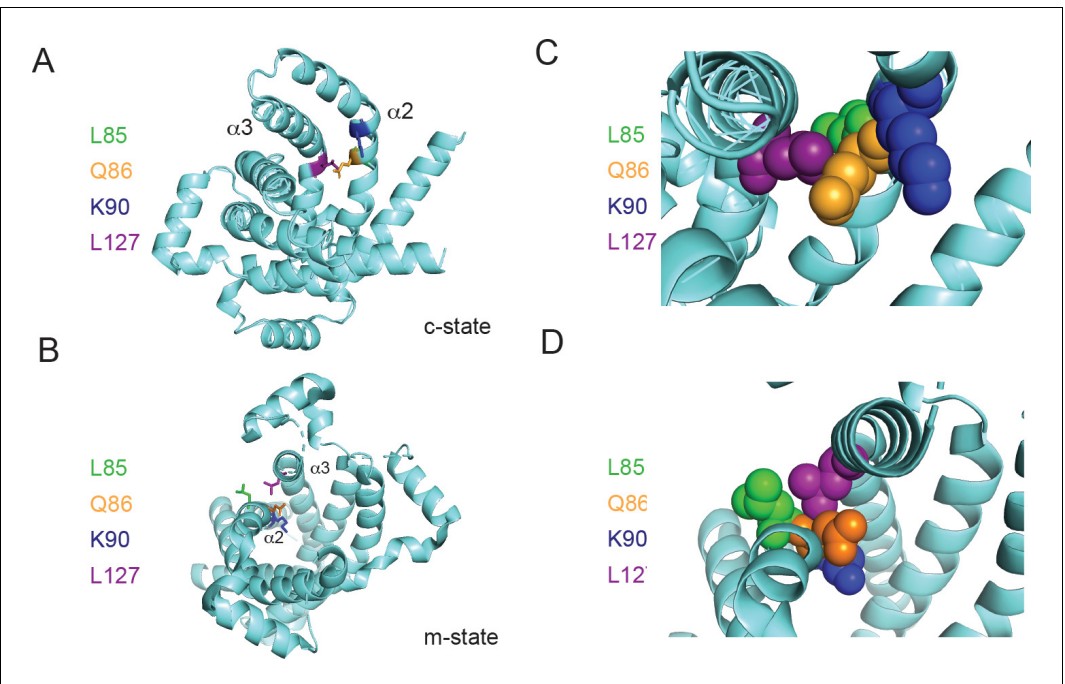

**Figure 8.** Positioning of Leu127 relative to adjacent residues on helix 2. Ribbon diagrams of PIC2 (**A**) c-state and (**B**) m-state structures. The polypeptide is shown as a ribbon trace (aquamarine) and the side chains as stick models. The Leu127 is colored purple to distinguish it from the adjacent Leu85 (green), Gln86 (orange), and Lys90 (blue) residues on helix 2 (α2). Enlargement of the Leu127 interaction with the surrounding residues shown as spheres in (**C**) c-state and (**D**) m-state.

The online version of this article includes the following figure supplement(s) for figure 8:

**Figure supplement 1.** NMR of the CuL and role of Y83 in interactions with the proposed C29-H33 binding site.

Our phylogenetic analysis revealed nine taxa that lack a PIC2-like ortholog but have retained COX. Each of these taxa have multiple *MIR1*-like transporters (*Guillardia theta, Thalassiosira pseudonana, Emiliania huxleyi, Dictyostelium discoideum, Ustilago maydis, Cyanidioschyzon merolae, Chrysochromulina tobinii, Micromonas commoda,* and *Naegleria gruberi*). Alignment of these paralogs identified residues that are present in at least one of the duplicates and are shared with PIC2 (*Figure 5* and *Figure 5—figure supplement 1*). We hypothesize that these variants may have allowed MIR1 to secondarily gain Cu transport activity. One consistent difference we observe is a histidine found in PIC2 orthologs versus a glutamine found in MIR1 orthologs at position 230 (as numbered in PIC2). Both of these side chains stabilize the conformation of a possible cardiolipin binding site by hydrogen bonding to peptide carbonyl oxygens. Additional experiments will be required to determine if this substitution affects substrate selectivity.

We favor an idea that *MIR1* duplication is a response to overcome the loss of PIC2 due to the basal polytomy among MCF subfamilies observed in our phylogenetic analyses of MCF proteins from each taxon. The lack of clear phylogenetic relationship to a second MCF group suggested that functional transitions are occurring within PIC2-MIR1 clades. However, this requires further investigation and an acknowledgement that other MCF transporters may have also acquired Cu transport activity. Indeed, in yeast we have shown that the MCF family member MRS3 serves as a secondary importer of mitochondrial Cu (*Vest et al., 2016*). MRS3 is known as an iron transporter, but transport of Cu by MRS3 and its orthologs has been reported in studies using mitochondrially derived vesicles from yeast and plants and in a reconstituted assay system (*Brazzolotto et al., 2014*; *Froschauer et al., 2009*; *Christenson et al., 2018*; *Jain et al., 2019*). MRS3 orthologs are not consistently recovered in a well-supported sister clade to the PIC2-MIR1 clade, suggesting that this functional redundancy is the result of convergent evolution.

## Understanding phosphate transport

Our biochemical data suggest that Lys90, Leu127, and Met275 are important for phosphate transport but are dispensable for Cu transport in *L. lactis*. The proposed mechanism of transport for MCFs based on the comparison of the c- and m-states of the ADP-ATP carrier suggests that even-numbered helices shift to allow transport/transition to the opposite state (*Ruprecht and Kunji, 2020*; *Ruprecht et al., 2019*). The PIC2 structural model shows that Leu127 is located on helix 3 adjacent to a proline that kinks the helix, thereby altering helix–helix packing interactions with helix 2 (*Figure 8*). The Leu127 side chain interacts with the peptide backbone between Leu85 (Met in SLC25A3) and Gln86 in a knobs-into-holes interaction. We hypothesize that helix 2 reorients in the alanine substitution mutant especially in the vicinity of Gln86, changing the dynamics of that part of the structure. In the c-state, this change could shift the side chains of Gln86, and therefore Lys90, to a conformation that disrupts a phosphate binding site (*Figure 8*) and, by extension, decreases its rate of transport. Methionine 275 is part of the computationally predicted conserved substrate contact point (*Robinson et al., 2008*). Based on its position in the c-state model below Lys90, Met275 is most likely involved in transport after phosphate enters deeper into the aqueous binding pocket of the protein (*Figure 4—figure supplement 2*).

Cu transport requires the formation of transient covalent bonds between the metal and ligands during transport, whereas phosphate transport relies on hydrogen bonding and salt bridges. These requirements may account for the fact that multiple mutations were able to inhibit the ability of PIC2 to transport phosphate. Other site-directed mutational studies of MIR1 have identified multiple residues that are required for phosphate transport (*Wohlrab et al., 2002*; *Briggs et al., 1999*; *Phelps et al., 1996*; *Wohlrab and Briggs, 1994*; *Phelps and Wohlrab, 1991*), including His33, Thr44, and Lys90 (using PIC2 numbering). Consistent with these earlier studies, we observe decreased phosphate transport when mutating the corresponding residues in PIC2. In fact, previous studies of MIR1 function showed that mutation of Thr44 to cysteine partially inactivated phosphate transport (*Phelps and Wohlrab, 1991*). This cysteine/threonine is clearly demarcated at the node between PIC2 and MIR1 clades, suggesting that it may be a critical change that weakened, but did not eliminate, phosphate transport in PIC2-like transporters (*Figure 5*). Three lineages (*Oryza sativa*, *Spizellomyces punctatus,* and *P. marinus*) lack MIR1-like transporters and have multiple PIC2-like transporters. In the case of rice, this could simply be due to the polyploid nature of its genome. In the chytrid *S. punctatus,* it may suggest that duplication enhances gene dosage. That is, additional copies compensate for less efficient phosphate transport. In contrast, the duplicated genes in *P. marinus* have undergone several notable changes; one variant has a large carboxy terminal truncation, three of the four variants have valine replacing cysteine at position 44 (as noted above from previous studies, threonine at this position is optimal for phosphate transport), and histidine at position 230 is replaced by the glutamine that is found in more phosphate-selective transporters. These changes and gene dosage may be sufficient to overcome the loss of a MIR1-like transporter. Testing these hypotheses will require in vitro expression of multiple transporters to assess substrate selection.

## Conclusions

Mitochondria function as a metabolic hub that controls physiology and disease by balancing the concentrations of multiple metabolites and essential elements (*Baker et al., 2017*; *Martínez-Reyes and Chandel, 2020*). The MCF proteins play a critical role in regulating the import and export of these substrates (*Cunningham and Rutter, 2020*; *Palmieri et al., 2020*), and have been duplicated and specialized over evolutionary time to selectivity recognize and transport highly similar substrates. However, gene duplication has allowed for the retention of some carriers with multiple substrates. The evolutionary relationships among these carriers reveal aspects of transport mechanisms and the physiological demands of the organism. Our analysis of the Cu–phosphate MCF transporters shows that organisms deploy multiple strategies to recruit these substrates. We cannot determine a single characteristic that indicates an advantage or disadvantage of either strategy as unique patterns appear nested in different lineages. Metal transport to the mitochondrial matrix is required for Fe–S cluster assembly and COX assembly. Perhaps metal substrates are sufficiently simple that multiple MCFs are capable of transport. However, given the fatal disorders that result from too much or too little Cu or iron, it is unlikely that their transport is left to chance (*Xu et al., 2013*). Storage in the

mitochondrial matrix may have evolved as a mechanism to ensure Cu availability for COX assembly in an early endosymbiont that was subsequently retained during eukaryogenesis (*Cobine et al., 2021*). Additional roles for Cu in the matrix remain to be determined. The recent discoveries that mitochondrial Cu can induce cell death through a pathway coined cuproptosis (*Tsvetkov et al., 2019*), disrupt essential processes such as Fe–S assembly (*Vallières et al., 2017*; *Brancaccio et al., 2017*), and alter the stability of SOD1 in the cytosol (*Boulet et al., 2018*) collectively suggest that understanding the physiological consequences of disrupting this Cu pool and its homoeostasis remains an important area of future research.

# Materials and methods

## Key resources table

| Reagent type (species) or resource | Designation | Source or reference | Identifiers | Additional information |
|---|---|---|---|---|
| Gene (*Saccharomyces cerevisiae*) | PIC2 | Saccharomyces Genome Database | SGD:S000000855 | |
| Gene (*Saccharomyces cerevisiae*) | MIR1 | Saccharomyces Genome Database | SGD:S000003838 | |
| Gene (*Mus musculus*) | SLC25A3 | doi:10.1074/jbc.RA117.000265 | Ensembl: ENSMUSG00000061904 | |
| Strain, strain background (*Saccharomyces cerevisiae*) | BY4741 | Thermo Scientific | | |
| Strain, strain background (*Lactococcus lactis*) | NZ9000 pepN::nisRK | doi:10.1007/s00253-005-0107-6 | | |
| Strain, strain background (*Escherichia coli*) | BL21(DE3) | New England Biolabs | Cat# C2527H | |
| Transfected construct (*Mus musculus*) | SLC25A3 | doi:10.1074/jbc.RA117.000265 | | |
| Cell line (*Mus musculus*) | MEF | doi:10.1074/jbc.RA117.000265 | | |
| Antibody | SLC25A3 (rabbit polyclonal) | This paper | | See 'Materials and methods' for details WB(1:2000) |
| Antibody | PIC2 (rabbit polyclonal) | doi:10.1074/jbc.M113.470674 | | WB(1:1000) |
| Antibody | COX1 (mouse monoclonal) | Abcam | ab14705 | WB(1:2000) |
| Antibody | TOM40 (rabbit polyclonal) | ProteinTech | 18409-1-AP | WB(1:2000) |
| Recombinant DNA reagent | EV | MoBiTec | Cat# ELV00200-01 | pNZ8148 expression vector containing the nisA promoter |
| Recombinant DNA reagent | WT; C21A; C29A; H33A; C44A; C225A; Q47A; V48A; Q86A; K90A; D124A; L127A; Q176A; G268A | This paper | See *Supplementary file 1* | pNZ8148 with wild-type PIC2 sequence or with individual residue mutated |
| Recombinant DNA reagent | pHis.Parallel.1 | doi:10.1006/prep.1998.1003 | | Expression vector for *E. coli* |
| Chemical compound, drug | Nisin | MoBiTec | Cat# VS-ELK01000-02 | |

*Continued on next page*

*Continued*

| Reagent type (species) or resource | Designation | Source or reference | Identifiers | Additional information |
|---|---|---|---|---|
| Chemical compound, drug | Silver | SPEX certiPrep | | Lot #19-13 AGX |
| Chemical compound, drug | Arsenate | Alfa Aesar | | Lot #U21A031 |
| Chemical compound, drug | Phen Green SK | Invitrogen | | Lot #2034143 |
| Software, algorithm | BioEdit | Ibis Biosciences, USA | RRID:SCR_007361 | http://www.mbio.ncsu.edu/BioEdit/bioedit.html |
| Software, algorithm | Coot | doi:10.1107/S0907444904019158 | RRID:SCR_014222 | |
| Software, algorithm | CD-HIT | doi:10.1093/bioinformatics/bts565 | RRID:SCR_007105 | |
| Software, algorithm | ConSurf server | doi:10.1093/nar/gkw408 | RRID:SCR_002320 | |
| Software, algorithm | Fiji (ImageJ) | doi:10.1038/nmeth.2019 | RRID:SCR_002285 | |
| Software, algorithm | HMMER, v3.3 | doi:10.1093/nar/gkr367 | RRID:SCR_005305 | http://hmmer.org |
| Software, algorithm | IQ-TREE, v2.0.3 | doi:10.1093/molbev/msu300 | RRID:SCR_017254 | http://www.iqtree.org/ |
| Software, algorithm | MEGA Software, v10.1.8 | doi:10.1093/molbev/msz312 | RRID:SCR_000667 | https://www.megasoftware.net/ |
| Software, algorithm | PHENIX | doi:10.1107/S2059798319011471 | RRID:SCR_014224 | https://www.phenix-online.org/ |
| Software, algorithm | Prism | Graph Pad, Inc | RRID:SCR_002798 | |
| Software, algorithm | PyMOL | Schrödinger, LLC | RRID:SCR_000305 | http://www.pymol.org/ |
| Software, algorithm | SWISS-MODEL | doi:10.1093/nar/gkg520 | RRID:SCR_018123 | https://swissmodel.expasy.org/ |
| Other | Bovine ADP/ATP carrier | doi:10.1038/nature02056 | PDB:1OKC | Crystal structure of the bovine ADP/ATP carrier |
| Other | Yeast ADP/ATP carrier, c-state | doi:10.1073/pnas.1320692111 | PDB:4C9G | Crystal structure of the yeast ADP/ATP carrier in the cytoplasmic-open state |
| Other | Yeast ADP/ATP carrier, m-state | doi:10.1016/j.cell.2018.11.025 | PDB:6GCI | Crystal structure of the yeast ADP/ATP carrier in the matrix-open state |

## Phylogenetic analysis

To delineate the evolutionary histories of the PIC2/MIR1 orthologs, 47 species were chosen that span the eukaryotic supergroups defined here. For each of these species, complete nuclear genome assemblies and protein predictions are available from NCBI (*Supplementary file 1*). MCF orthologs were identified using HMMER (*Potter et al., 2018*) to detect sequences containing the mitochondrial carrier (MC) domain (PFAM PF00153). Redundant sequences and transcript variants were eliminated using CD-Hit with a threshold of 0.9 (*Huang et al., 2010*).

To distinguish PIC2/MIR1 orthologs from other members of the MCF family, phylogenetic trees were built using the MC domain containing proteins from each organism as well as the complete set of MCF proteins from *Homo sapiens* and *S. cerevisiae*. Amino acid sequences were aligned in MEGA X (*Kumar et al., 2018*) using ClustalW with default parameters. Neighbor joining trees were generated using a Poisson substitution model, uniform substitution rates among sites, and pairwise gap deletion. Support values were determined using 1000 bootstrap replicates.

Amino acid sequences of the eukaryotic MIR1/PIC2 orthologs were aligned with 32 *S. cerevisiae* MCF proteins using MUSCLE implemented in MEGA X. Phylogenetic analysis was performed using IQ-TREE version 2.0.3 (*Minh et al., 2020*). The optimal substitution model was selected using the IQ-TREE ModelFinder (*Kalyaanamoorthy et al., 2017*). A maximum likelihood tree was constructed using the LG+F+R7 model (a general codon exchange matrix for nuclear genes with amino acid frequencies determined empirically from the data and seven rate categories). Support was calculated based on 1000 replications using ultrafast bootstrap approximation (UFBoot2; *Hoang et al., 2018*).

## Structural modeling

Multiple sequence alignments between PIC2, its mammalian orthologs, and the ADP/ATP exchanger family were used to correctly place indels at secondary structure boundaries between helices and loops on the molecular surface and also to ensure robust alignment of the key helices. Insertions and deletions were modest in the resultant alignment (three deletions and one insertion, ranging from 5 to 10 residues each; *Supplementary file 1*). Initial molecular models of the cytosolic open form of PIC2 were generated using Swissmodel (*Waterhouse et al., 2018*) from the atomic coordinates of the yeast ATP/ADP exchanger bound to carboxyatractyloside (PDB:4C9G) (*Ruprecht et al., 2014*). The aligned PIC2 and yeast ATP/ADP carrier sequences share 19% identity and 37% similarity over 305 residues. The resultant model was carefully compared to the parent structure and to the structure of the cytosolic open form of the bovine mitochondrial ATP/ADP carrier bound to carboxyatractyloside (PDB:1OKC) (*Pebay-Peyroula et al., 2003*), again 19% identical and 37% similar to PIC2 over 305 residues. Side chains in the model were adjusted for sensible hydrogen bonding, salt bridge formation, and consistency and packing of rotamers using Coot (*Emsley and Cowtan, 2004*). Finally, the model atomic coordinates were energy minimized within the PHENIX suite to optimize molecular geometry and relieve steric clashes (*Liebschner et al., 2019*). Using the atomic coordinates of the bongkrekic acid-bound matrix open form of the yeast ATP/ADP carrier (PDB:6GCI) (*Ruprecht et al., 2019*), we then constructed a model of the matrix open form of yeast PIC2. Both models appeared to be sensible, preserving secondary structure, cardiolipin-binding sites, and exhibiting the expected constellation of hydrophobic residues facing the membrane. Using the same methodology, we also built atomic models of the cytosolic open and matrix open forms of mouse SLC25A3 (also 18% identical and 37% similar over 305 residues to the template structures), and then carefully compared the PIC2 and SLC25A3 models for consistency of helix positions, side chain packing, and hydrogen bonding (PIC2 and SLC25A3 sequences are 47% identical and 64% similar over 312 residues with only 8% indels).

## Expression in *L. lactis*

*L. lactis* cells transformed with vector (pNZ8148 [MoBiTec]) alone or pNZ8148 carrying yeast MIR1, PIC2, or site-directed PIC2 mutants were grown overnight at 30°C in M17 medium with 0.5% glucose and 10 µg/mL chloramphenicol. To determine $Ag^+$ toxicity, cells were grown in a 96-well plate containing M17 medium plus 1 ng/mL nisin and increasing concentrations of $Ag^+$ (0–250 µM) or $AsO_4^{3-}$ (0–2.5 mM). Controls containing M17 without nisin or M17 plus $Ag^+$ or $AsO_4^{3-}$ without nisin were included. Optical density at 600 nm was used to assess growth after 24 hr. Percent growth was quantified by comparing to the optical density of the same genotype in nisin alone.

## Elemental analysis

Samples were digested in 40% nitric acid by boiling for 1 hr in capped, acid washed tubes, diluted in ultra-pure, metal-free water and analyzed by ICP-OES (Perkin Elmer, Optima 7300DV) versus acid-washed blanks. Concentrations were determined from a standard curve constructed with serial dilutions of two commercially available mixed metal standards (Optima). Blanks of nitric acid with and without 'metal-spikes' were analyzed to ensure reproducibility.

## Cell lines

Clonal *Slc25a3*$^{Flox/Flox}$ and *Slc25a3*$^{-/-}$ MEF lines (*Boulet et al., 2018*) were maintained at the University of Saskatchewan and verified as *Mycoplasma*-free using the MycoAlert Mycoplasma Detection Kit (Lonza). The integrity of the mutation was routinely verified using PCR and immunoblotting.

## Cell culture conditions

Clonal $Slc25a3^{Flox/Flox}$ and $Slc25a3^{-/-}$ MEF lines were cultured in high-glucose DMEM (Dulbecco's Modified Eagle's medium) containing sodium pyruvate, 50 μg/mL uridine, 0.1 mM mercaptoethanol, and 10% fetal bovine serum at 37°C at an atmosphere of 5% $CO_2$ (*Boulet et al., 2018*). Mouse *Slc25a3-b* cDNA was amplified from RNA and cloned into a Gateway-modified retroviral expression vector. The fidelity of this construct was confirmed by Sanger sequencing and retrovirus was produced with the Phoenix Amphotrophic packaging cell line used to transduce MEFs.

## Immunoblot and activity assays

This study used monoclonal antibodies raised against COX1 (Abcam ab14734), a rabbit polyclonal antibody TOM40 (ProteinTech 18409-1-AP), and a rabbit polyclonal antibody raised against the KLH conjugated SLC25A3 peptide CRMQVDPQKYKGIFNGSVTLKED (Pacific Immunology). For *L. lactis* extracts, we used rabbit polyclonal antibody raised against a PIC2 peptide (*Vest et al., 2013*). COX activity was determined by monitoring the decrease in absorbance at 550 nm of chemically reduced cytochrome *c* in the presence of whole cell or mitochondrial extracts (*Cobine et al., 2004*). All activities were normalized to protein concentration, then converted to percentage of maximum control value.

## Expression of recombinant proteins

PIC2 and MIR1 from *S. cerevisiae* (strain BY4741) were sub-cloned into pHis parallel 1 for *Escherichia coli* expression. BL21(DE3) *E. coli* transformed with the vector were grown to optical density at 600 nm of 0.6–0.8, and protein expression was induced with isopropyl β-D-1-thiogalactopyranoside for 2 hr. Inclusion bodies containing the recombinant proteins were isolated as described (*Palmieri et al., 2000*). Cells were resuspended in cell lysis buffer (150 mM NaCl, 50 mM Tris pH 7.5) and disrupted by sonication. Insoluble material was collected by centrifugation at 18,500 *g* and resuspended in cell lysis buffer and loaded onto a stepwise 40%, 53%, 70% sucrose gradient. Samples were centrifuged at 18,500 *g* for 1 hr, and inclusion bodies were isolated at the interface of the 53% and 70% layers. Recombinant proteins were solubilized in 6 M urea, then incorporated into liposomes by mixing egg yolk phospholipids before overnight dialysis in 25 mM Tris buffer pH 7.2. The dialyzed mixture was sonicated in the presence of the metal-responsive fluorophore Phen Green, then purified by loading the vesicles in 35% sucrose under 20% sucrose and centrifuged at 18,500 *g* for 60 min. The final proteoliposomes were isolated from the top of the sucrose layer and protein concentration was determined by Bradford assay.

## Purification and NMR analysis of CuL

Intact mitochondria were fractionated in soluble and insoluble fractions as described previously (*Vest et al., 2016*). The soluble anionic fractions were isolated by adding DEAE (Whatman) resin in batch. The resin was washed with 20 mM ammonium acetate, pH 8.0, and eluted with of 1 M ammonium acetate, pH 8.0. The samples were dried and loaded onto a Phenomenex C18 column. Unbound fractions were removed with 50 mM ammonium acetate, pH 5.0 followed by a 60 min gradient to 100% acetonitrile. The final fractions were analyzed for copper by ICP-OES (PerkinElmer, Optima 7300DV) and for fluorescence (PerkinElmer Life Sciences LS55 fluorimeter). Excitation and emission scans of copper-containing fractions used an excitation maximum of 320 nm and an emission maximum of 400 nm with 5-nm slit widths. For NMR analysis, samples were dried in CentriVap concentrator (Labconco), then resuspended in 100% $D_2O$. The process of drying and resuspending in $D_2O$ was repeated at least five times placing samples in >95% $D_2O$. Samples were analyzed on a 500-mHz NMR spectrometer (Varian, Inova) referenced to residual solvent. $^1H$ 1D spectrum, $^{13}C$ 1D spectrum, $^1H^{13}C$ HSQC, and $^1H^{13}C$ HMBC experiments were collected. Purity of individual samples could not be confirmed by mass spectroscopy based on failure to reliably detect ions. However, the identical $^1H$ and $^{13}C$ spectra were consistently produced from the final product of the chromatography.

## Fluorescence anisotropy

Purified CuL isolated from mitochondria was diluted in 150 mM NaCl, 50 mM Tris pH 7.5 to give a fluorescence intensity (excitation at 320 nm and emissions at 400 nm) of 30–50. MCF proteins

incorporated into liposomes were added in 1–5 µL increments, and anisotropy was measured using a PerkinElmer Life Sciences LS55 fluorescence spectrometer.

## Acknowledgements

We thank Auburn University undergraduate research students Ann Ashton, Shannon Baseke, Shelley Brannon, LaQuasha Jones, Kacie Oglesby, Christina Waples, and Mary Wetzel for contributing to construction and initial testing of the different mutants used in this study. This work was supported by a grant from the National Institutes of Health (R01GM120211 to PAC and SCL). PAC is supported by the Alabama Agricultural Experiment Station. KMB is supported by a grant from the National Science Foundation (EF 2021886).

## Additional information

### Funding

| Funder | Grant reference number | Author |
|---|---|---|
| National Institutes of Health | R01GM120211 | Scot C Leary<br>Paul A Cobine |
| National Science Foundation | EF 2021886 | Katherine M Buckley |
| Alabama Agricultural Experiment Station | | Paul A Cobine |

The funders had no role in study design, data collection and interpretation, or the decision to submit the work for publication.

### Author contributions

Xinyu Zhu, Paul A Cobine, Conceptualization, Data curation, Formal analysis, Supervision, Funding acquisition, Validation, Investigation, Visualization, Methodology, Writing - original draft, Project administration, Writing - review and editing; Aren Boulet, Formal analysis, Investigation, Visualization, Writing - review and editing; Katherine M Buckley, Casey B Phillips, Conceptualization, Data curation, Formal analysis, Validation, Investigation, Visualization, Methodology, Writing - original draft, Writing - review and editing; Micah G Gammon, Laura E Oldfather, Investigation; Stanley A Moore, Investigation, Visualization, Writing - original draft, Writing - review and editing; Scot C Leary, Conceptualization, Formal analysis, Funding acquisition, Validation, Investigation, Visualization, Writing - original draft, Project administration, Writing - review and editing

### Author ORCIDs

Xinyu Zhu https://orcid.org/0000-0002-7618-1501
Katherine M Buckley https://orcid.org/0000-0002-6585-8943
Scot C Leary https://orcid.org/0000-0001-8488-7822
Paul A Cobine https://orcid.org/0000-0001-6012-0985

### Decision letter and Author response

Decision letter https://doi.org/10.7554/eLife.64690.sa1
Author response https://doi.org/10.7554/eLife.64690.sa2

## Additional files

### Supplementary files

• Supplementary file 1. The accession numbers of the genome sequences, accession number of all mitochondrial carrier family-containing transcripts, PIC2/MIR1 sequences, the sequences removed by CD-HIT 0.9 threshold, the comparison of PIC2 and MIR1 entropy scores and conservation of residues, and multiple sequence alignment used for structural modeling of PIC2 and c-state model of PIC2.

• Transparent reporting form

### Data availability

All data generated or analyzed during this study are included in the manuscript, supplementary file, and available on GenBank.

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
