## [Decision Letter]

**Acceptance summary:**

This work synthesizes bioinformatics, in vivo, and in vitro transport assays to understand the molecular basis for substrate selection and promiscuity of the mitochondrial carrier family (SLC25). This comprehensive work provides insight into mitochondrial physiology and the evolutionary dynamics that have contributed to mitochondrial transporter specificity.

**Decision letter after peer review:**

Thank you for submitting your article "Mitochondrial copper and phosphate transporter specificity was defined early in the evolution of eukaryotes" for consideration by *eLife*. Your article has been reviewed by three peer reviewers, including Randy B Stockbridge as the Reviewing Editor and Reviewer #1, and the evaluation has been overseen by Olga Boudker as the Senior Editor. The following individual involved in review of your submission has agreed to reveal their identity: Steven Michael Claypool (Reviewer #2).

Below you will find a set of recommendations, which, if completed, would lead to publication in *eLife*.

Essential revisions:

1) To fairly compare the growth phenotypes/transport capacity of the assorted mutants relative to WT Pic2, immunoblots should be included documenting their relative expression.

2) The reviewers had the following comments on improving the presentation of the bioinformatics:

– During the bioinformatic process, the authors used CD-Hit 0.9 to reduce the redundancy. Did this process cause removal of sequences that are supposed to present in Figures 2 and 3? If they were removed, is there any justification to remove those genes (these >90% identity genes represent recent gene duplications)? Or did the authors not remove any of the redundancy in Figures 2 and 3? This should be clarified.

– Figure 2 showed that three sequences fell outside of PIC2-MIR1 clusters. Why is this happening? Is it because the first NJ tree clustering included some sequences from other clusters? If so, does this mean that the NJ clustering may missed some PIC2-MIR1 sequences by clustering with other sequences?

– What are sequences that are neighboring cluster to PIC2-MIR1 clusters? How can the authors ensure that those neighboring sequences are not functionally homologous (promiscuous Cu and/or phosphate transport) to PIC1/MIR1? This ambiguity comes from the lack of presentation of the entire tree of the superfamily. If the neighbor cluster exhibit relatively close sequences to PIC2/MIR1, the authors should reconsider their hypothesis and perform further analysis. If the neighbour sequence cluster is very distant (and the function of the clusters are known to be different from Cu and phosphate), then the authors can argue that the functional transitions are occurring only within PIC2/MIR1 families.

3) The manuscript would be improved with additional background information on the Ag+ uptake assay. The reviewers had a number of questions, including: why does the *pic2* yeast have increased sensitivity to low amounts of the copper mimic, Ag+? Some additional insight as to why the absence of the mitochondrial Copper transporter would compromise respiratory growth would be helpful. In the absence of Pic2, does Ag+ accumulate in cytoplasm and this is more toxic than if it were to be transported into the matrix? Why can Ag+ be a mimic of Cu? Does PIC2 only carry Cu+ (not Cu2+)? Does that mean that any positively charged transition metals are transported by PIC2? If so, why is PIC2 called a Cu+ transporter specifically?

4) Is there any possibility that other transporters beside PIC2/MIR1 are involved in either pathway? In particular, other homologous sequences in the MCF superfamily may be playing a role?

5) The M275A mutant should be included in the discussion of "Understanding phosphate transport." Are K90 and M275 conserved in SLC25A3? According to the homology models, is the proposed role of M275 in phosphate transport distinct from K90 and L127?

6) The discussion of CuL was confusing to a non-specialist. It seems very speculative that the intact CuL complex might be transported, especially since this ligand is not present in the lactis/silver system that is set up as a proxy for mitochondrial copper transport. If the copper were to remain bound to a high affinity CuL ligand, wouldn't that undercut the hypothesized importance of copper-liganding sidechains such as cys or his?

7) The molecular-level analysis about van der Waals contacts and sidechain rotamers (related to Figure 8 and Figure 8—figure supplement 1) probably goes too far for a homology model. This should be softened somewhat, and for this more detailed analysis, it might be nice to consider whether alternative models show similar sidechain arrangements.

---

## [Author Response]

Essential revisions:1) To fairly compare the growth phenotypes/transport capacity of the assorted mutants relative to WT Pic2, immunoblots should be included documenting their relative expression.

We have added Western blot data to show that all mutant proteins are expressed in *Lactococcus lactis*. These data have been added to Figure 6 as panel A, and establish that the steady-state levels of all variants save for Q47A and G268A are comparable to wild-type PIC2. We have altered the Discussion to mention the relative levels of protein expressed.

2) The reviewers had the following comments on improving the presentation of the bioinformatics:– During the bioinformatic process, the authors used CD-hit0.9 to reduce the redundancy. Did this process cause removal of sequences that are supposed to present in Figures 2 and 3? If they were removed, is there any justification to remove those genes (these >90% identity genes represent recent gene duplications)? Or did the authors not remove any of the redundancy in Figures 2 and 3? This should be clarified.

One of the biggest challenges with working with this type of data is that the quality of genome assemblies varies greatly among taxa. High-quality genome sequences based on extensive transcriptomic data are available for commonly used model systems (e.g., humans, *Drosophila*, *Arabidopsis*) while high-depth, quality data are more sparse for other lineages, resulting in more fractured genome assemblies. To address the reviewer’s concern regarding removed sequences, we added a table to Supplementary file 1 of the sequences that were removed and have added details highlighting the considerations for eliminating or retaining the sequences in the table and to the figure legend that describes our bioinformatics pipeline (Figure 2—figure supplement 1). We believe that the strategy is sound as the MCF sequences we retrieved from human transcriptomic datasets include 312 transcripts derived from 54 MCF genes, and at the 90% threshold we retained 85 of these transcripts which cover 52 of 54 MCF genes. Similar results were obtained in other model systems using this threshold: in *Arabidopsis*, 60 transcripts were used in our analysis of 58 known MCF genes; in *Drosophila* we used 52 transcripts from 48 MCF genes. When using the lower-quality genome assemblies that are available for many of the more distant species, it is difficult or impossible to differentiate closely related paralogs and transcript isoforms from allelic copies or genome assembly artifacts. However, for the non-model species (i.e. everything except human, *Drosophila*, zebrafish, and *Arabidopsis*), 94% of transcripts were retained at the 90% threshold, suggesting that very few potential transcripts of interest were eliminated. The transcripts that were eliminated are now listed in Supplementary file 1.

– Figure 2 showed that three sequences fell outside of PIC2-MIR1 clusters. Why is this happening? Is it because the first NJ tree clustering included some sequences from other clusters? If so, does this mean that the NJ clustering may missed some PIC2-MIR1 sequences by clustering with other sequences?

In the bioinformatic pipeline, the initial groupings were determined by creating a neighbor-joining tree for all presumed MCFs (at least 2 MCF domains defined by PFAM PF00153) in that taxa, combined with the human and yeast MCFs (Figure 2—figure supplement 1 step 2). In these neighbor-joining trees the sequences present in a cluster, with MIR1/PIC2/SLC25A3 being retained for further analysis (see example in Figure 2—figure supplement 3). From the 47 taxa analyzed only three had sequences that group in the maximum likelihood tree with PIC2/MIR1 and were retained as “other”. Of these three, the *Acanthamoeba castellanii* sequence appears to be the most divergent in the grouping as it has only 2 MCF domains and is missing the first 60 amino acids of PIC2/MIR1 orthologs, as well as key residues such as C29 H33 Q86 K90. Therefore, we speculate that this duplication may be pseudogenizing. Because this sequence is included in our analysis, we believe that the level of stringency of the pipeline is calibrated to avoid the loss of important gene duplicates. The functions of these “other” MCFs neighboring the PIC2-MIR1 clade remain unknown. While it is clear that these sequences are more closely related to the PIC2-MIR1 clade than any other group of MCF sequences, biochemical assays of transport would be required to establish that these are bona fide PIC2-MIR1 orthologs.

– What are sequences that are neighboring cluster to PIC2-MIR1 clusters? How can the authors ensure that those neighboring sequences are not functionally homologous (promiscuous Cu and/or phosphate transport) to PIC1/MIR1? This ambiguity comes from the lack of presentation of the entire tree of the superfamily. If the neighbor cluster exhibit relatively close sequences to PIC2/MIR1, the authors should reconsider their hypothesis and perform further analysis. If the neighbour sequence cluster is very distant (and the function of the clusters are known to be different from Cu and phosphate), then the authors can argue that the functional transitions are occurring only within PIC2/MIR1 families.

The topology of the 44 individual NJ trees consisting of all MCFs from specific taxa and the complete set of yeast and human MCF proteins does not allow us to confidently infer a closest MCF group to PIC2/MIR1. The deep evolutionary relationships, short sequences and small number of “required” residues to adopt the MCF domains results in a polytomy for the different MCF types. This lack of consistent neighbor led us to favor a hypothesis that the functional transitions are only occurring within PIC2/MIR1 families. We intend to pursue biochemical data for the MIR1 duplications present in taxa without PIC2 in future studies.

3) The manuscript would be improved with additional background information on the Ag+ uptake assay. The reviewers had a number of questions, including: why does the pic2 yeast have increased sensitivity to low amounts of the copper mimic, Ag+? Some additional insight as to why the absence of the mitochondrial Copper transporter would compromise respiratory growth would be helpful. In the absence of Pic2, does Ag+ accumulate in cytoplasm and this is more toxic than if it were to be transported into the matrix? Why can Ag+ be a mimic of Cu? Does PIC2 only carry Cu+ (not Cu2+)? Does that mean that any positively charged transition metals are transported by PIC2? If so, why is PIC2 called a Cu+ transporter specifically?

Silver (Ag) has been used extensively as an isoelectronic mimetic of cuprous Cu. We have added a description and references to support that in the Introduction as well as extended the introductory materials to highlight how we have leveraged Ag to interrogate PIC2 function (Vest et al., 2013). The fact that it can be a Cu(I) mimetic is exploited here in two distinct ways: 1) as a supplement in yeast growth media to severely limit total available Cu, and 2) as an effective antimicrobial for Cu transport assays using *L. lactis* strains expressing PIC2 variants.

In yeast, supplementing growth media with increasing Ag concentrations results in decreased levels of total cellular Cu and causes a specific decrease in mitochondrial Cu levels at non-toxic concentrations (Vest et al., 2013). We demonstrated that this decrease in Cu in mitochondria results from Ag acting as a competitive inhibitor of Cu uptake. This suggested that Ag could be exploited to induce a COX defect. We have shown that yeast lacking *PIC2* and *MRS3* become COX deficient at lower concentrations of Ag, suggesting that these strains had a decreased ability to recruit Cu (Vest et al., 2013, Vest et al., 2016). Therefore, in yeast, the addition of Ag blocks COX assembly by limiting mitochondrial Cu availability. In *L. lactis*, we exploited the toxicity of Ag, hypothesizing that expression of PIC2 in the cytoplasmic membrane would lead to increased Ag accumulation and result in growth arrest. Indeed, the presence of a Cu/Ag transporter in the membrane effectively shifts the minimal inhibitory Ag concentration and causes a more severe growth arrest. These assays collectively suggest that PIC2 is required to overcome Ag induced Cu deficiency in yeast and is responsible for the uptake of Ag and increased toxicity in *L. lactis*.

The redox state of Cu transported by PIC2 has been established using purified proteins reconstituted into proteoliposomes and heterologously expressed proteins in the *L. lactis* system. Multiple experiments by us and others have confirmed the majority of Cu in mitochondria is Cu(I), with redox specific chelators, EPR, X-ray absorption spectroscopy all failing to detect appreciable amounts of Cu(II). These established datasets and the fact that Ag(I) acts as a competitive inhibitor of Cu uptake strongly suggest that Cu(I) is the substrate of PIC2. In experiments in Vest et al., 2013 with *L. lactis*, we added Cu(II) but *L. lactis* constitutively produces menaquinone which acts as an endogenous reductant. Thus, it is likely that *L. lactis* was transporting Cu(I). While Cu(II) can be added to proteoliposomes, the chelator Phen Green cannot discriminate redox state so the evidence for Cu(II) transport is less clear. Additionally we saw no evidence for directly for transport of other metals tested (Ca, Zn, Mn, Ni, Fe) into liposomes or as competitors for Cu transport. Importantly, our expectation was that Cu in the form of CuL would be the major transported form of the metal ion. However, the transporter is clearly able to transport both ionic Cu and CuL. We have expanded on the experimental evidence that supports PIC2’s ability to transport Cu(I)/CuL, and further discussed the implications of this finding for the transport mechanism.

4) Is there any possibility that other transporters beside PIC2/MIR1 are involved in either pathway? In particular, other homologous sequences in the MCF superfamily may be playing a role?

Deletion of *PIC2* and *MRS3* in yeast does not completely eliminate mitochondrial Cu accumulation, suggesting that additional transporters or some other uptake mechanism must exist. It was our hope that by expanding our knowledge of PIC2 paralogs in diverse taxa we would identify such transporters. However, the phylogenetic signal from the neighbor joining trees of all MCFs in these taxa showed different “types” of MCFs as the closest relatives, and these adjacent branches show low support. Our future goals will build off this study to test the Cu transport activity of duplicated *MIR1* sequences that are found in the absence of PIC2 orthologs.

5) The M275A mutant should be included in the discussion of "Understanding phosphate transport." Are K90 and M275 conserved in SLC25A3? According to the homology models, is the proposed role of M275 in phosphate transport distinct from K90 and L127?

We thank the reviewers for pointing out this oversight. The M275A mutant shows normal Ag transport but is inhibited for AsO_4_^3-^ transport. The position of M275 in the aqueous binding pocket of the transporter in the model suggests it would act as a contact point after initial binding of phosphate to K90 and Q86. This binding could enhance the transition to the matrix open state, allowing for transport of the substrate. We have discussed the potential significance of this mutant to our understanding of substrate transport in the revised manuscript.

6) The discussion of CuL was confusing to a non-specialist. It seems very speculative that the intact CuL complex might be transported, especially since this ligand is not present in the lactis/silver system that is set up as a proxy for mitochondrial copper transport. If the copper were to remain bound to a high affinity CuL ligand, wouldn't that undercut the hypothesized importance of copper-liganding sidechains such as cys or his?

Multiple studies use in vitro transport and fluorescence anisotropy assays to show that the transported substrate for PIC2 is the CuL. However, the biochemical assays we performed here suggest PIC2 is also capable of transporting ionic Cu into mitochondria. We significantly expanded the Discussion in the manuscript to provide a more detailed explanation of the historical perspective on mitochondrial Cu delivery as well as the major unknowns in this pathway. In addition to explaining the proposed roles for CuL, we have expanded on the roles that cysteine and histidine residues would play in transport of either ionic Cu or the CuL. Aromatics, such as the ring within the CuL identified by NMR and fluorescence spectroscopy, are known to have favorable SH/π interactions within proteins. Therefore, the CuL could be interacting with this pair of residues. We have also added more details to clarify the potential mechanisms of transporting the intact CuL versus release of Cu to the transporter.

7) The molecular-level analysis about van der Waals contacts and sidechain rotamers (related to Figure 8 and Figure 8—figure supplement 1) probably goes too far for a homology model. This should be softened somewhat, and for this more detailed analysis, it might be nice to consider whether alternative models show similar sidechain arrangements.

We have edited this section to generalize the interpretations that we make based on the model we have selected.